# Graphene-based sensing of oxygen transport through pulmonary membranes

Mijung Kim [1,3], Marilyn Porras-Gomez [2,3] & Cecilia Leal [2✉]

Lipid-protein complexes are the basis of pulmonary surfactants covering the respiratory surface and mediating gas exchange in lungs. Cardiolipin is a mitochondrial lipid over-expressed in mammalian lungs infected by bacterial pneumonia. In addition, increased oxygen supply (hyperoxia) is a pathological factor also critical in bacterial pneumonia. In this paper we fabricate a micrometer-size graphene-based sensor to measure oxygen permeation through pulmonary membranes. Combining oxygen sensing, X-ray scattering, and Atomic Force Microscopy, we show that mammalian pulmonary membranes suffer a structural transformation induced by cardiolipin. We observe that cardiolipin promotes the formation of periodic protein–free inter–membrane contacts with rhombohedral symmetry. Membrane contacts, or stalks, promote a significant increase in oxygen gas permeation which may bear significance for alveoli gas exchange imbalance in pneumonia.

[1] Department of Electrical and Computer Engineering, University of Illinois at Urbana-Champaign, Urbana, IL 61801, USA. [2] Department of Materials Science and Engineering, University of Illinois at Urbana-Champaign, Urbana, IL 61801, USA. [3]These authors contributed equally: Mijung Kim, Marilyn Porras-Gomez. ✉email: cecilial@illinois.edu

The exchange of carbon dioxide and oxygen in the lungs is mediated by alveolar membranes which comprise three main layers: (i) the epithelial basement membrane, (ii) the alveolar epithelium, and (iii) the fluid containing a surfactant layer. In the alveoli, this surfactant layer interfaces the gas phase and the alveolar fluid lining (hypophase), a complex mixture of macrophages and type II cells that secrete several multiscale lipid–protein aggregates. Pulmonary surfactant receives its name because its basic function is to lower the surface tension at the alveoli air/aqueous interface. The composition of lung surfactant varies throughout vertebrates but it is generally composed of several lipids (approximately 90%) and proteins (approximately 10%)[1,2] organized in a variety of hierarchic assemblies such as vesicles, multilamellar bodies, and tubular myelin[3,4]. More specifically, there is a mixture of phospholipids, where zwitterionic phosphatidylcholines predominate (40–70 wt% mostly dipalmitoylphosphatidylcholine—such as 1,2-dipalmitoyl-sn-glycero-3-phosphocholine, DPPC), anionic lipids such as phosphatidylglycerol (in particular 1,2-dioleoyl-sn-glycero3-phospho-(1′-rac-glycerol) (sodium salt), DOPG), neutral lipids (like cholesterol at 8–10 wt%), and surfactant proteins (SP)[1,5–7]. The function of this complex pulmonary membrane goes beyond lowering the surface tension[3] providing the first barrier against oxygen transport from the alveoli into the bloodstream[8,9]. However, its role, in particular in pathological contexts, remains elusive.

Perez-Gil and co-workers[7,10,11] showed that lung membrane structure could be responsible for facilitating oxygen diffusion through the air–water interface. A minimal density of surfactant is required for oxygen diffusion rates to increase compared to that in surfactant-free water layers. The postulated mechanism was that hydrophobic surfactant proteins SP-B and SP-C stabilize the formation of membrane contacts and that these proteo-lipid channels promote membrane permeability and small molecule diffusion through the alveolar surfaces[12,13].

It has been established that increased oxygen supply in the lungs (hyperoxia) is a pathological factor in bacterial pneumonia. This has been observed in newborn mice where hyperoxia potentiates bacterial growth and inflammatory responses[14], as well as being an important co–factor for the development of acute lung injury and lethality in *L. pneumophila* pneumonia[15]. In addition, lung fluid from humans and mice infected with bacterial pneumonia displays significantly elevated contents of cardiolipin, a mitochondrial-specific phospholipid[16]. We have recently demonstrated that cardiolipin induces significant water loss in lung membrane extracts and model multilamellar bodies[17].

Cardiolipin appears to be involved in different transport processes, in particular that of calcium ions ($Ca^{2+}$)[18]. Calcium can induce phase changes in lipid membranes containing negatively charged phospholipids such as cardiolipin[19,20]. In addition, elevated calcium levels in the lungs of cystic fibrosis patients facilitate chronic behavior of *P. aeruginosa*[21].

It is noteworthy that many studies on lung surfactants, including those to formulate surfactant–replacement therapies, treat the system as monolayers or bilayers of lipids and proteins highly diluted on a pristine air/water interface. However, the alveolar fluid is a crowded environment comprising multiple cell types and large lipid–protein aggregates where there is not a lot of free water. The determination of surface tension is highly dependent on surface stiffness[22]. Hence, the modulation of surface stress by the lung surfactant will differ when it is adsorbed onto air/dilute-water versus air/viscous-fluid interfaces. The latter recapitulates more accurately the alveolar air/fluid interface and it might be beneficial to study lung surfactant properties in a regime of 100% relative humidity (RH) instead of bulk free water. Using a theoretical framework, the groups of Sparr and Wennerström demonstrated that tubular lipid aggregates similar to tubular

myelin at 100% RH enhance transport of oxygen when compared to stacks of planar bilayers[8,23].

Various types of oxygen gas sensors including electrochemical[24,25], optical[10,26,27] and field-effect transistor (FETs)-based ones[28–30] have been studied for many different applications. For instance, Olmeda et al.[10] developed oxygen optical microsensors made of an oxygen-sensitive luminescent probe, and evaluated the kinetics of oxygen diffusion through dilute lung membrane systems[10]. A solid-supported lipid membrane platform using FETs has been shown to be one of the most useful model systems for the investigation of the properties and functions of thin biological membranes[31,32]. Sensing based on FETs is advantageous because of miniaturization, low cost, fast response time, and high sensitivity[28,33]. For oxygen gas sensing this includes graphene-based devices[34–37], carbon nanotubes[30,38,39] and oxide semiconductors such as $In_2O_3$, and $TiO_2$[40–42]. Among them, graphene is one of the best materials for measuring oxygen gas transport through lipid membranes since it is robust, ultraflat, biocompatible, and stable in high-humidity environments. Additionally, its two-dimensional structure endows it with a large surface-to-volume ratio and great sensitivity enhancement. Finally, one of the main advantages of this sensor is that solutions can be deposited as thin films directly onto the device, allowing the measurement of oxygen permeation in combination with structural characterization.

We develop a basic science framework to generate biophysical insights of the structure of lung membranes and how that impacts oxygen flow. We design graphene-based FET oxygen sensing devices that are amenable to the incorporation of supported lung membranes in a high-humidity environment. We probe oxygen transport in bovine lipid–protein extract surfactant and lipid-only model systems in conditions mimicking a healthy lung membrane and comparing them against one afflicted by bacterial pneumonia.

We combine oxygen sensing with structural characterization techniques such as X-ray scattering in grazing incidence (GISAXS), confocal microscopy, and atomic force microscopy (AFM). Our main finding is that cardiolipin in the presence of calcium ions promote the formation of abundant and periodic inter-membrane stalks. Inter-membrane contacts will occur independent of SP content. Under these conditions, oxygen permeation is significantly enhanced. Akin to proteo-lipid channels, the hydrophobic intermembrane stalks facilitate oxygen gas transport through the alveolar membranes. We postulate that in pathological contexts like pneumonia lung membranes experience significant structural changes that directly impact oxygen permeation.

## Results and discussion

**Oxygen gas sensor fabrication and characteristics.** To investigate the relation between lung surfactant structure and oxygen transport, we fabricated a FET-based oxygen gas sensor onto a silicon oxide ($SiO_2$)/silicon (Si) wafer. Multilamellar lipid membranes were deposited directly onto the device (Fig. 1a), which consists of graphene as the sensing material (biocompatible and amenable to high humidity), $SiO_2$ as the dielectric layer, silicon as the gate electrode and chromium and gold as source/drain electrodes. Supplementary Fig. 1 shows the schematics of the electrodes and sensing arrays. There is a total of 50 sensors in one device, and both the length and width of the graphene channels are 50 μm (Fig. 1b). The micrometer size of the channel, which is the effective sensing area, is compliant with variable degrees of pulmonary membrane coverage. However, a uniform membrane with very few defects and no dewetting instabilities is preferable. Fluorescence microscopy images (Fig. 1c and Supplementary Fig. 2) show that the films display sufficient device coverage. The

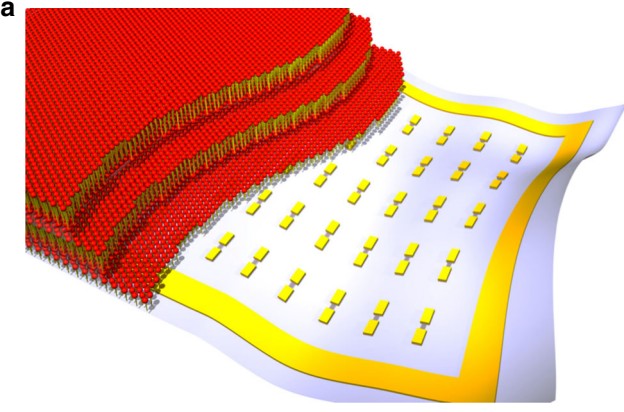

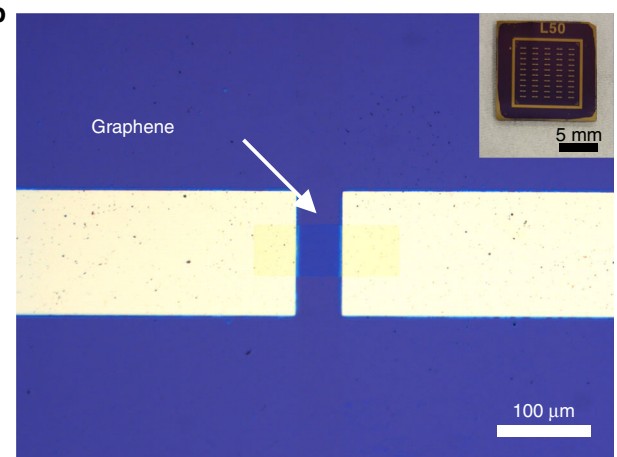

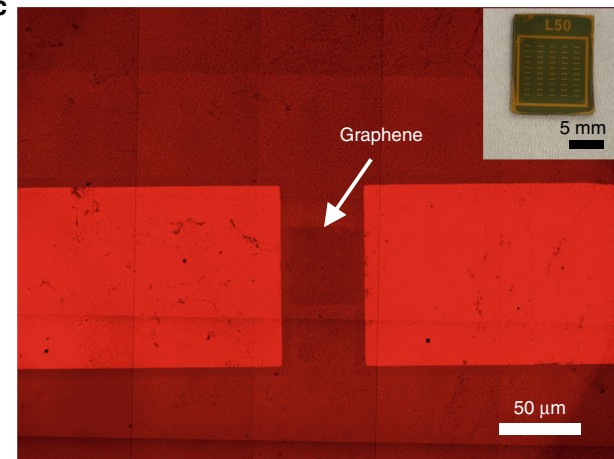

**Fig. 1 Sensing platform for oxygen transport through pulmonary membranes. a** Schematic illustration of lipid membranes deposited on the FET-based sensor which serves as a solid support for stacked lipid bilayers. **b** Bright field optical microscopy image of the oxygen gas sensor. The white arrow indicates the graphene channel between gold electrodes. Channel length and width are 50 μm. Inset: photograph of the sensor which has total 50 FETs on a 1 cm × 1 cm SiO₂/Si substrate. **c** Fluorescence microscopy image of fluorescently labeled lipid membranes coated on the FET sensor. Graphene is marked with a white arrow. Inset: photograph of the sensor coated with lipid films.

lipid films in this example are composed of a model pulmonary membrane with a 3:1 DPPC:DOPG molar ratio. Changing the composition of the lipid film does not compromise sample coverage (Supplementary Fig. 2).

The mechanism of oxygen sensing is that in the FET sensor, the applied gate voltage induces a surface potential, thereby modulating channel conductance (Fig. 2a). Any binding events altering the surface charge or the surface potential will be detected due to changes in the electrostatic potential. When oxygen gas molecules diffuse and adhere to the graphene surface they act as a p-type dopant since an oxygen atom in $O_2$ molecules is an electron-withdrawing group[43]. The electrons in the channel are attracted and transferred to $O_2$ molecules, producing holes in the conduction band of graphene. Therefore, the attached $O_2$ molecules on the graphene will enhance hole conduction and generate a significant decrease in resistance.

The transfer characteristics of the graphene FET sensor before and after deposition of lung lipid membranes (Fig. 2b) shows that, without lipids, the graphene FET has a Dirac voltage ($V_{Dirac}$) of ~2 V with similar electron and hole conductance. However, the graphene FET coated with lipid membranes exhibits a shift in $V_{Dirac}$ to (~2.4 V) as well as a considerable decrease in electron and hole mobility (slope of $V_g$–$I_d$ curve). $V_{Dirac}$ shifts to higher voltage due to a charge-impurity potential caused by the presence of negatively charged lipids in lung membranes (DOPG)[29,44]. In addition, the reduction in electron and hole mobility stems from the increase in total scattering explained by the coupling between charge carriers and lipid membranes.

We measured sensing responses ($I/I_0$) by altering the ratio of oxygen gas ($O_2$) to oxygen and nitrogen gases ($O_2 + N_2$) by varying their volume (ranging from 0 to 80%) (Fig. 2c). The responses are defined by the ratio of device currents with the flow of both nitrogen gas and oxygen gas ($I$) to the currents with the flow of only nitrogen gas ($I_0$). The measured response data (dots) and their corresponding fitted curves (lines) indicate outstanding linearity of response to the analyte (oxygen gas) with a correlation coefficient ($R^2$) of 0.998, both with and without lipids, suggesting a valid sensing performance.

**Thickness and morphology of pulmonary membranes**. We fabricated gas sensors to measure oxygen permeability through lung membranes directly deposited onto the sensing device. In this respect, it is imperative to obtain uniform lung membrane film thickness. We have shown that lipid film thickness, even for complex mixtures, can be controlled by spin-coating speed[45–47]. The thickness and morphology of lung membranes can be determined by AFM using the tip-scratch method (Fig. 3). We probed film thickness in both lipid-only model systems and bovine lipid–protein surfactant extracts for conditions mimicking healthy and diseased states. The diseased state consisted of a lung membrane system with elevated concentrations of cardiolipin in the presence of $Ca^{2+}$. The samples were prepared by spin-coating the stock solutions directly onto the FET sensor. We kept the total lipid concentration constant and added cardiolipin and $Ca^{2+}$ as molar percentage (mol%) of the total lipids. The spin-coating time and rate is fixed at 30 s and 4000 rpm, respectively. To obtain accurate film thickness results via the AFM tip-scratch method, only the film must be scratched while ensuring that the substrate suffers no damage. A force of 3 μN was found to scratch only the films but not the device surface. Intermittent contact mode scans confirmed this fact. Measured depths through cross-section analysis of scratches is indicative of film thickness. AFM 3D height data of healthy and diseased lung surfactant membranes are shown in Fig. 3, in particular, AFM height images of the healthy model system (Fig. 3a) composed of 3:1 DPPC:DOPG, and the diseased model system (Fig. 3b) consisting of 3:1 DPPC:DOPG with cardiolipin (8 mol% of total lipids) and $Ca^{2+}$ (4 mol% is equivalent to 1 mM), respectively. The thickness of these two membranes was comparable (~100 nm) as shown

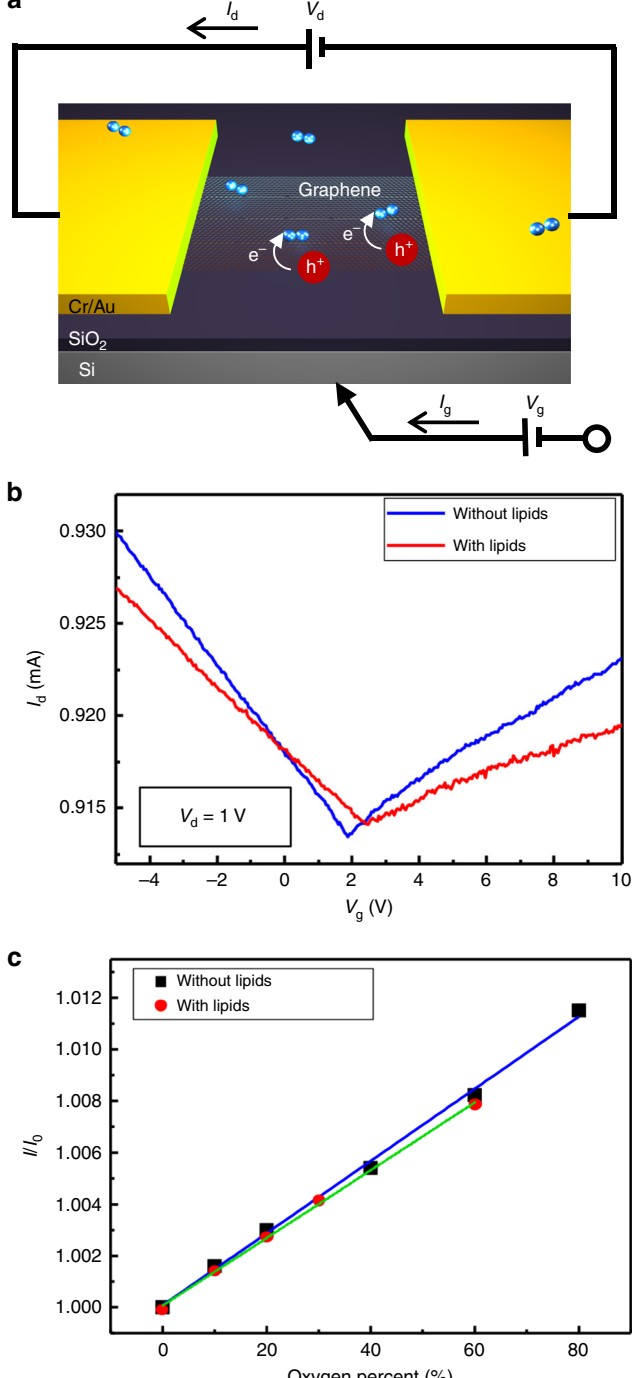

**Fig. 2 Electric characteristics of graphene FET sensors. a** Schematic image describing how the FET-based sensor detects the oxygen gas using graphene. Black arrows represent the direction of drain current ($I_d$). **b** Transfer characteristics of graphene FET. Drain voltage ($V_d$) is 1 V and gate voltage ($V_g$) is swiped from −5 to 10 V. Blue and red curve indicate $I_d$–$V_g$ curves of the FET sensor without and with lipid films, respectively. **c** Sensing response calibration graph of the oxygen gas sensor as function of oxygen percent from 0 to 80% with and without lipids. Dots represent the real response data. Blue and green lines are linear fits of with and without lipids, respectively (correlation coefficient ($R^2$) = 0.998).

in the cross-section topography profiles. Compared to healthy mimetic membranes, diseased-mimetic membranes exhibit higher surface roughness, consistent with cardiolipin and $Ca^{2+}$-induced morphological changes[17].

The AFM images of lung extract samples in healthy and diseased states are shown correspondingly in Fig. 3c–d. Healthy lung extract samples have a thickness of ∼60 nm with a rather homogeneous smooth surface. However, the diseased lung extract exhibits rough topography and comprises two parts: peaks with the thickness of ∼110 nm and valleys with a thickness of ∼60 nm. Similar to the results obtained for the model system, cardiolipin in the presence of $Ca^{2+}$ appear to enhance surface roughness in lung extract samples. Supplementary Fig. 3 provides additional analysis on the thickness and the morphology of lung-mimetic membranes with different compositions (3:1 DPPC:DOPG with only cardiolipin at 8 mol% and only $Ca^{2+}$ at 4 mol%). We will discuss further AFM data in a later section.

**Cardiolipin affects membrane structure and oxygen transport.** To investigate the effect of cardiolipin and $Ca^{2+}$ on the structure and associated oxygen transport of pulmonary membranes, we employed GISAXS analysis of lung membranes directly adsorbed onto the graphene oxygen sensor device. GISAXS was used for structural studies in conjunction with oxygen transport measurements at full humidity (98–100% RH) conditions. The two-dimensional GISAXS data of all samples exhibit a series of equally spaced diffraction peaks significantly more intense along the $q_z$ direction (orthogonal to the surface) compared to along the $q_{xy}$ direction (parallel to the surface), indicating that lung membranes are arranged in a multilamellar array mostly aligned with the $q_z$ direction normal to the sample surface. Diffraction along the $q_z$ direction ranging from 0 to 0.1 $Å^{-1}$ are due to the graphene FET sensor (Supplementary Fig. 4).

Higher resolution synchrotron GISAXS data for a 3:1 DPPC/DOPG membrane with 4 mol% $Ca^{2+}$ and 8 mol% cardiolipin (Fig. 4a) reveals a series of periodic reflections in $q_z$ at constant $q_{xy}$ ∼ ±0.05 $Å^{-1}$ indicated by white arrows. This diffraction pattern is characteristic of a rhombohedral bilayer stack structure, often referred to as a stalk phase. This consists of stacks of bilayers with periodic inter-bilayer contacts (Fig. 4b). This stalk phase formed by bilayer hemifusion "point deffects" has been identified as an intermediate structure between discrete and fully fused membrane systems[48–51]. The stalks are formed by merging monolayers in close contact whereas distal bilayers remain discrete. The formation of this structure requires that lipid building blocks form monolayers that can tolerate local negative curvature domains[49]. $Ca^{2+}$ has been implicated in promoting negative curvature and overall membrane fusion in a number of model and natural lipid systems[52,53]. In addition, cardiolipin, especially when combined with calcium ions, has been shown to induce phase transitions of single bilayer vesicles into monolayer systems with high negative mean curvature[54]. Comparing the GISAXS diffraction patterns of healthy (Fig. 4c) membranes with the diseased states containing $Ca^{2+}$ (Fig. 4d) one can observe the emergence of additional diffraction spots at around $q_{xy}$ ∼ ±0.05 $Å^{-1}$ indicated by white arrows. Adding cardiolipin to the system at increasing concentration (Fig. 4e–f) intensifies the diffraction spots.

We have previously demonstrated[17] that the combination of cardiolipin and calcium ions promotes in-plane domain formation in model lung membrane systems (here represented schematically by the green and light blue areas in Fig. 4b). In that work, we postulated that diseased lung membranes might have inter-bilayer contacts that are protein-independent; however, the stalk phase is hard to detect in bulk systems where membrane microcrystallites are randomly oriented in three dimensions. This is because the diffraction spots arising from the rhombohedral stalks would

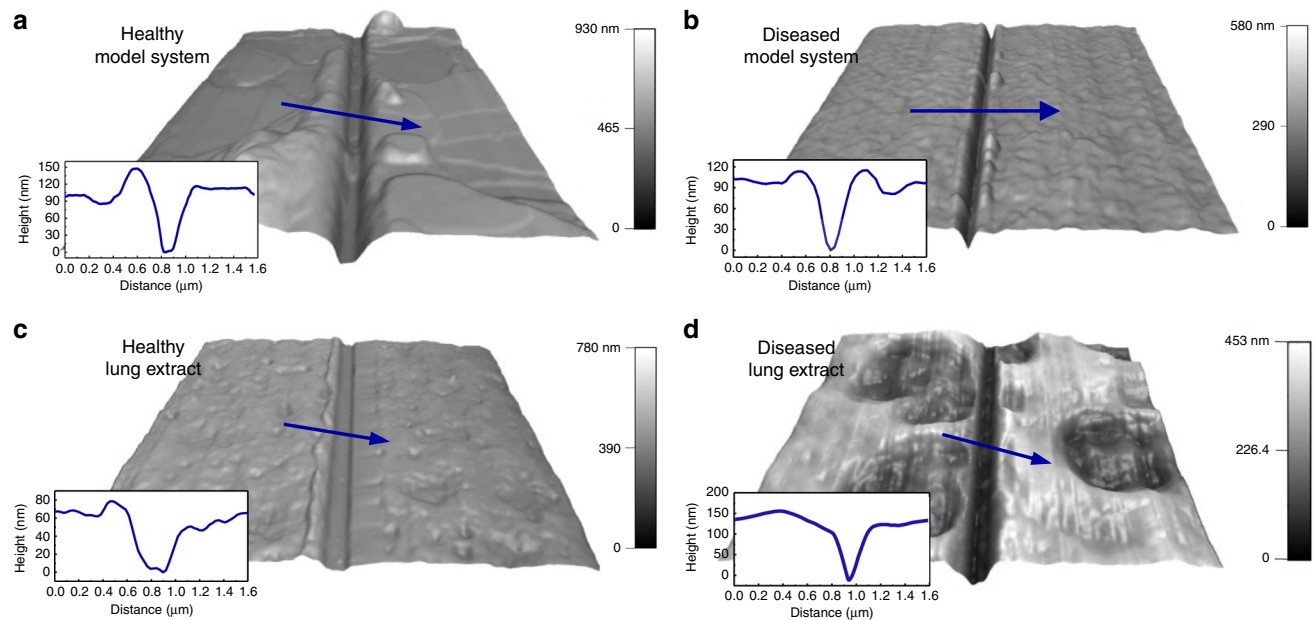

**Fig. 3 Film thickness of model systems and lung extracts.** AFM 3D height images and cross-sectional profiles of **a** healthy model systems composed of 3:1 DPPC:DOPG, scan size 4 μm × 4 μm, **b** diseased model system consisting of 3:1 DPPC:DOPG with addition of cardiolipin and Ca$^{2+}$, scan size 8 μm × 8 μm, **c** healthy bovine lipid–protein extract surfactant (BLES), scan size 11 μm x 11 μm, and **d** diseased BLES with cardiolipin and Ca$^{2+}$. Blue arrows display scanning directions of profiles, scan size 10 μm × 10 μm.

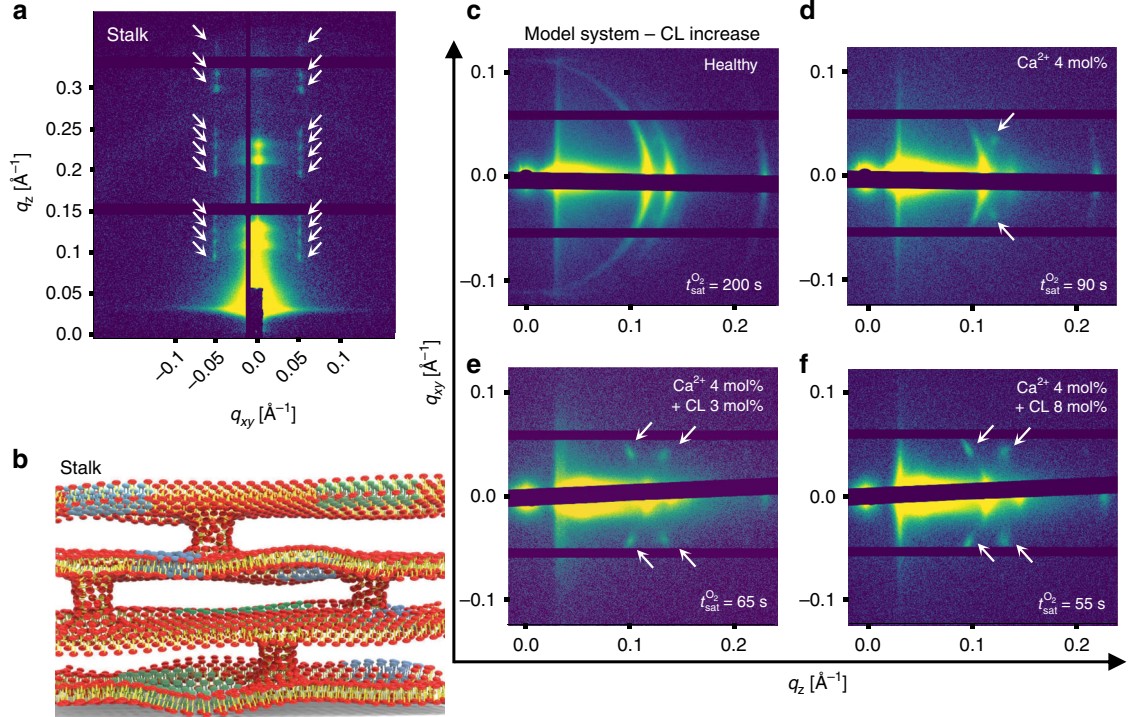

**Fig. 4 GISAXS of model diseased pulmonary membranes. a** Synchrotron GISAXS of the diseased lung membrane model system. White arrows indicate stalk phase peaks which appear along the $q_z$ direction at $q_{xy} \sim \pm0.05$ Å$^{-1}$. **b** Schematic illustration of a stalk phase consisting of spatially ordered inter-bilayer contacts across a membrane with four stacked bilayers. Blue and green areas represent possible in-plane domains. **c–f** In-house GISAXS diffraction patterns of healthy model systems (**c**) and model systems with Ca$^{2+}$ 4 mol% (**d**), with Ca$^{2+}$ 4 mol% + cardiolipin (CL) 3 mol% (**e**) and with Ca$^{2+}$ 4 mol% + CL 8 mol% (**f**). Stalk-phase peaks are indicated by white arrows. Oxygen gas saturation times ($t_{sat}^{O_2}$) depicted in the GISAXS images corresponds to the time that FET sensors take to get saturated with a flow of 20% oxygen gas.

appear as diffraction rings that can coincide with those of the lamellar phase. In this paper, the fact that there is a strong alignment of the bilayer stacks parallel to the device surface, permitted the detection of the rhombohedral membrane structure which is formed by spatially correlated bilayer contacts along $q_z$. Additional 2D GISAXS patterns of model lung membrane systems at fixed cardiolipin and increasing calcium ion content are provided in Supplementary Fig. 5.

**Table 1 Oxygen gas saturation times ($t_{sat}^{O_2}$) through pulmonary membranes.**

| Sample | Model system $t_{sat}^{O_2}$ | BLES $t_{sat}^{O_2}$ |
|---|---|---|
| Healthy 3:1 DPPC:DOPG | 200 | 50 |
| +[Ca$^{2+}$ 4 mol%] | 90 | 47 |
| +[Ca$^{2+}$ 4 mol% + CL 3 mol%] | 65 | 38 |
| +[Ca$^{2+}$ 4 mol% + CL 8 mol%] | 55 | 33 |

$t_{sat}^{O_2}$ of control sample, healthy and diseased such as +[Ca$^{2+}$ 4 mol%], +[Ca$^{2+}$ 4 mol% + CL 3 mol%] and +[Ca$^{2+}$ 4 mol% + CL 8 mol%] for model systems and BLES, respectively. Bear device control $t_{sat}^{O_2}$ = 20 s.

Oxygen permeation through lung membranes at different conditions was measured on the exact same samples used for GISAXS structural determination. We probed oxygen gas saturation time ($t_{sat}^{O_2}$), which denotes the required time for the oxygen sensor to be saturated upon exposure to gas with 20% oxygen. Hence $t_{sat}^{O_2}$ is inversely proportional to the oxygen permeability through pulmonary membranes. We measured (Table 1) the oxygen gas saturation time (in seconds) for the lipid-only lung membrane model and the bovine lipid extract surfactant (BLES) systems at fixed Ca$^{2+}$ and increasing cardiolipin concentration (raw FET sensor data are provided in Supplementary Figs. 6 and 7). The control sample is the oxygen gas sensor with no adsorbed membranes. Saturation times for BLES are considerably shorter than those of the model systems due to two reasons: (1) lung extract membranes spin-coated onto the sensor at the same conditions as the lipid-only model membranes are slightly thinner as described in Fig. 3, and (2) the lung system in its healthy state already exhibits inter-membrane contacts (or hemifusions) that are stabilized by lung proteins SP-B and SP-C[10,55]. Hemifusion enhances oxygen gas permeability by generating a favorable pathway for small hydrophobic oxygen molecules. This mechanism will be discussed in detail later in the manuscript.

The lipid-only, lung-mimetic healthy membrane (with cardiolipin 0 mol% and Ca$^{2+}$ 0 mol%) with multilamellar peaks along the $q_z$ direction and without any stalk peaks (Fig. 4c) displays $t_{sat}^{O_2}$ of approximately 200 s. When Ca$^{2+}$ (4 mol%, equivalent to 1 mM) is added to the system (Fig. 4d) two weak stalk peaks emerge (white arrows) and $t_{sat}^{O_2}$ decreases to 90 s. It is noteworthy that the formation of hydrophobic contacts across bilayers in the multilamellar aggregate are promoted without the need of SPs and the impact on oxygen permeation is very significant. After adding 3 mol% of cardiolipin, four stalk diffraction spots appear and become more intense, indicative of more well-ordered interbilayer contacts. With the increase in cardiolipin concentration, the stalk patterns become stronger and the $t_{sat}^{O_2}$ keep decreasing to 55 s which is of the order of that observed in lung membrane extracts ($t_{sat}^{O_2}$ = 50 s) where inter-bilayer contacts are naturally stabilized by SPs. As in the model system, oxygen saturation times of the lipid–protein extract surfactant system decreases with addition of Ca$^{2+}$ and cardiolipin. Strikingly, the addition of cardiolipin 8 mol% and Ca$^{2+}$ 4 mol% leads to a reduction in the permeation time of oxygen from 50 to 33 s. This suggests that cardiolipin and Ca$^{2+}$ induce the formation of inter-membrane hydrophobic contacts in BLES additionally to those naturally existing and mediated by SP-B and SP-C. We do not know the form of the stalks or membrane contacts when SPs are present but the data suggests that they might form in addition to those stabilized by SPs. The stalk phase characteristic GISAXS patterns could not be clearly detected in the BLES samples (Supplementary Fig. 8). It is also clear from the 2D GISAXS data that the

orientation of the BLES multilayers with respect to the device surface is not as strong as what is observed in model systems. The stalk phase diffraction peaks will only emerge in nearly perfectly oriented membranes. If membranes are misaligned the rhombohedral diffraction spots will appear as rings that can coincide with the rings of the lamellar periodicity. In addition, protein-mediated stalks do not necessarily have rhombohedral periodicity. Hence, stalk diffraction peaks are not expected to be detectable in BLES membranes in healthy or diseased states. The fact that oxygen permeation is significantly enhanced is a good indication of additional membranes stalks. We argue that a membrane system with high density of hydrophobic membrane contacts will be more prone to oxygen permeation. As a control experiment we measured the oxygen permeation through a canonical lipid material known to form bicontinuous cubic phases. Bicontinuous cubic phases (Q$_{II}$) are highly porous comprising a continuum hydrophobic membrane intercalated by continuous water channels. Glycerol monooleate (GMO), and 1,2-dioleoyl-3-trimethyl-ammonium-propane chloride salt (DOTAP) at 85:15 molar ratios form bicontinuous cubic structures of the gyroid type (Q$_{II}^G$) in the form of lipid films when equilibrated at 100% RH[45]. As shown in the X-ray diffraction pattern (Supplementary Fig. 9), GMO/DOTAP at a 85:15 molar ratio adsorbed onto the oxygen sensor device yields peaks that are completely consistent with a (Q$_{II}^G$) phase with the gyroid (Ia3d) symmetry. In this system, the measured oxygen permeation is strikingly similar ($t_{sat}^{O_2}$ = 62 s) to that obtained for the stalk phase in the diseased-mimetic systems of our work. This is somewhat expected as stalk phases are related to bicontinuous phases and are often intermediate (yet thermodynamically stable)[56] structures between lamellar and (Q$_{II}$) phases.

Figure 5a–b shows 1D GISAXS profiles (Intensity versus $q$) after integration of the 2D GISAXS diffraction patterns on a narrow area along $q_z$ for the model and the lung extract systems in healthy (a) and diseased (b) states 5c–d. Additional 1D GISAXS profiles obtained for model and BLES lung membrane systems at different compositions are presented in Supplementary Fig. 10. Also in Fig. 5 AFM images in height and phase mode (e, f) for the lipid-only model membranes and the lung extracts (g, h) are shown. In the healthy state, both the model membrane and BLES display clear lamellar GISAXS peaks (indicated as 001, and 002) with a repeat distance $d = 2\pi/q_{001} = 98$ Å. Consistent with our previous findings in the same systems in bulk, a second set of lamellar peaks (marked by 001′, $d_{001'} = 80$ Å) emerges due to domain formation within lung membranes and alignment of domains in registry, which is expected for this lipid mixture of saturated and unsaturated phospholipids[17,57]. In the diseased state of the model system, clear lamellar peaks are still present in the integrated profiles. Note that the 1D profiles are integrated in a narrow area such that stalk-phase peaks are only visible in the 2D diffraction pattern (Fig. 4) and do not interfere with the 1D multilamellar peaks. Taken together, the 1D and 2D GISAXS data reveal the existence of well-oriented lamellar phases and periodic inter-bilayer stalks generated by Ca$^{2+}$ and cardiolipin (Fig. 5d). In contrast, the diseased lung extract films display considerably weaker lamellar peaks. This differs to the results obtained for bulk BLES systems in diseased states[17] where clear lamellar phases could be detected. This indicates that indeed BLES samples are not as well oriented with respect to the device substrate when compared to the model system. Hence, it is not surprising that no stalk-phase diffraction patterns are observed in 2D GISAXS of the BLES samples in healthy or diseased states (Supplementary Fig. 8). It is noteworthy that membrane misalignment across the air/water interface in the alveoli and increased stalk formation caused by the presence cardiolipin and Ca$^{2+}$ are both expected to

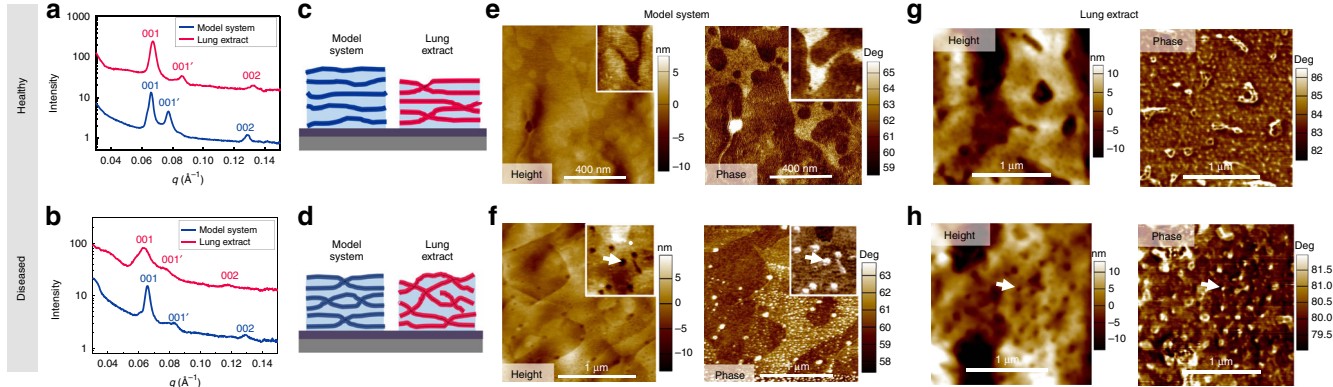

**Fig. 5 1D GISAXS and AFM of model systems and lung extracts.** 1D GISAXS intensity versus reciprocal space $q$ ($\text{Å}^{-1}$) profiles for BLES samples and model membrane systems in healthy (**a**) and diseased (**b**) states. The middle panels are schematic representation of stalks and orientation in healthy (**c**) and diseased (**d**) model (blue) and BLES (magenta) membrane systems. The space between bilayers (light blue) represents water layers. AFM height and phase images of healthy (**e**) and diseased (**f**) model membrane systems are represented in the figure. Inset in the healthy model (**e**) is 250 nm × 250 nm while inset in the diseased model (**f**) highlighting the pores in the diseased model system is 500 nm × 500 nm, white arrows point to pore-like defects. Height and phase AFM images of lung extract (BLES) in healthy (**g**) and diseased (**h**) states are also displayed.

induce significant oxygen permeation imbalance. Irregularities in $O_2$ and $CO_2$ gas transport and exchange might lead to abnormalities in blood gas composition which is a factor implicated in lung diseases[14,15]. It could be argued that phase separation in the lung membranes could be implicated in oxygen permeability. This is a possibility; however, the oxygen permeation through a control sample with higher degree of phase-separation (4:1 DPPC/DOPG)[17] yielded slightly lower oxygen permeation ($t_{sat}^{O_2} = 215\,\text{s}$) compared to the parent 3:1 DPPC/DOPG sample ($t_{sat}^{O_2} = 200\,\text{s}$) (Supplementary Fig. 6). Prior AFM studies have effectively captured that natural lung surfactants and lipid–based models experience hemifusion and membrane defects processes mediated by the addition of SPs[58–62].

AFM methods, in particular in phase-mode are very convenient to improve contrast in membrane systems. Changes in the phase lag reflect changes in the materials characteristics such as mechanical properties. Membrane defects like domain formation and enhancement of inter-bilayer contacts can possibly be observed using this technique. AFM imaging (Fig. 5e–f and Supplementary Fig. 11) was used to study protein-free lipid-based healthy (Fig. 5e) and diseased (Fig. 5f) membrane models as well as BLES (Fig. 5g–h). The morphology (in the height image) and materials properties (in the phase image) of lung extract membranes is significantly altered in the diseased state. Membrane stalks or hemifusion regions will appear as perforation-like defects, readily visible in the height images (marked by white arrows) which are correlated with brighter spots (higher angle) in the phase images, indicating that membrane pore-like defects have relatively higher elasticity than the surrounding membrane. In addition, these defects appear across the entire scanned area and vary in size ranging from a few nanometers to a 100 nm in diameter.

In this work we fabricated a FET-based sensor using graphene as the sensing material to measure oxygen gas permeation of thin (approximately 100 nm) lung membranes directly spin-coated onto the device. Employing graphene-based oxygen sensing, X-ray scattering, and AFM we discovered that lipid-only model systems comprising stacks of bilayers have much lower oxygen permeation capacity when compared to lipid–protein extract surfactants. This is consistent with the fact that SPs stabilize inter-bilayer contacts which act as hydrophobic pathways to enhance oxygen transport across multilamellar lung membranes. Surprisingly, when lung membranes are disturbed by bacterial pneumonia lipids (such cardiolipin) and calcium ions, a set of protein-free inter-bilayer contacts, or stalks, are formed. In a model system, the stalks are spatially ordered with rhombohedral symmetry. The oxygen permeation through a membrane with stalks induced by pneumonia lipids is significantly increased, even for a mammalian lung extract membrane comprising pre-existing protein-stabilized bilayer contacts.

The mechanism of enhanced oxygen gas transport in stalk membrane phases is schematically represented in Fig. 6. When oxygen is transported through lipid membranes, it spends most of the time bypassing water layers due to $O_2$ non-polarity and low water solubility. In contrast, $O_2$ is expected to move relatively fast through the hydrocarbon tails within the bilayers. Stalks, which are apolar hydrophobic contacts between bilayers, improve the permeability of non-polar oxygen gas through the membranes via the newly formed hydrophobic pathways, reducing the exposure of $O_2$ to the water phase (Supplementary Fig. 12 shows a zoomed-in schematic representation of a single membrane stalk).

The healthy lipid-only model system is composed of DPPC (Fig. 6 represented in red) and DOPG (represented in green) at a 3:1 molar ratio, which is considered a good model-system for pulmonary surfactants. The lipids self-assemble onto a hard graphene device surface in well-aligned lamellar structures composed of bilayers separated by water layers and no inter-bilayer contacts. This system shows the lowest oxygen gas permeability among all the membrane types. The diseased model system contains cardiolipin (represented in yellow) and calcium ions (purple) in 3:1 DPPC:DOPG membranes which is relevant for lungs afflicted with bacterial pneumonia[16]. Cardiolipin and $Ca^{2+}$ promote the formation of protein-free hydrophobic inter-bilayer contacts (stalks) that are periodic (rhombohedral phase). The stalk-phase activates transport pathways for oxygen gas, boosting the overall $O_2$ permeability through a diseased lung membrane model system. Lipid–protein extract surfactants comprise lipids (dark green) and SPs. In the healthy state, membranes exhibit randomly distributed inter-bilayer contacts which are stabilized by surfactant proteins SP-B (blue) and SP-C (red). Hence, lung extract membranes are more permeable to oxygen gas compared to a lipid-only model systems, which agrees with other studies that show accelerated oxygen diffusion due to protein-promoted inter-membrane connections in multilayered surfactant assemblies[10]. Cardiolipin induces the formation of periodic hemifusion contacts in lipid membranes, in particular in the presence of $Ca^{2+}$. It is likely that cardiolipin, which is overexpressed in

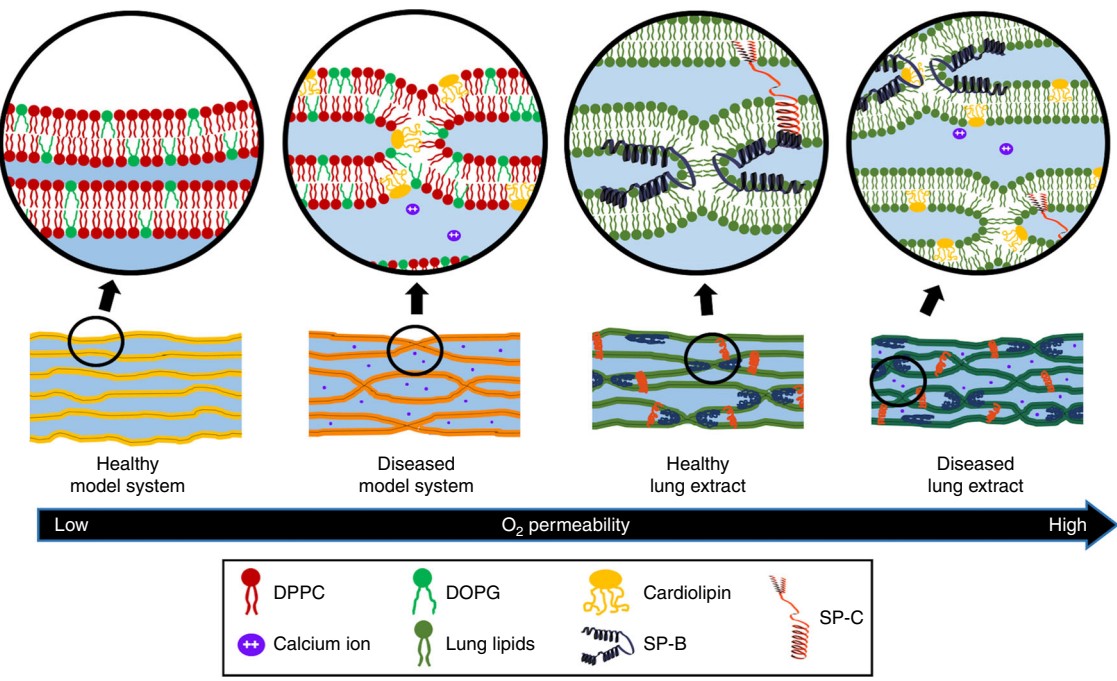

**Fig. 6 Mechanism of enhanced oxygen gas transport through stalk phases.** Schematic illustrations of model systems and lung extracts in healthy and diseased states and how their different structures affect oxygen gas permeability. Enlarged illustrations (circles) describe detailed structures of each system, displaying lipids (e.g., DPPC, DOPG, Cardiolipin, and BLES), surfactant proteins (SP-B and SP-C) and calcium ions.

diseased lungs, promotes the formation of such stalks between bilayers of lung membranes in the alveoli. The formation of membrane stalks leads to enhanced oxygen gas permeability through the hydrophobic hemifusion points. Respiratory pathologies often cause edema and fluid flow imbalance in the alveoli and an interruption of oxygen flow should be expected, however, abnormally accelerated oxygen gas transport (hyperoxia) has been observed in mammalian lungs in diseased states, in particular during pneumonia[14,15]. We conjecture that this is consistent with a defective lung membrane with an increased number of stalk defects. The next stage of underpinning the role of cardiolipin and enhanced oxygen permeation during pneumonia will require the structural characterization and oxygen permeability measurements of lung membranes extracted from healthy and diseased mammalian lungs. At this point, our results raise fundamental and conceptual insights on lung membrane function, indicating that changes in structure and composition directly relate to oxygen permeation. This observation can potentially enable innovative ideas for clinical research on the role of unbalanced oxygen diffusion through lung membranes in a pathological context.

## Methods

**Materials**. For sensor preparation, poly(methyl methacrylate) (950 PMMA A2) was purchased from MicroChem Corp (Westborough, MA) and graphene/copper foil was purchased from ACS Material, LLC (Pasadena, CA). $SiO_2$/Si wafers were purchased from UniversityWafer, Inc. (South Boston MA). For lipid film preparation, DPPC (1,2-dipalmitoyl-sn-glycero-3-phosphocholine) 25 g/L, CL (cardiolipin, heart, bovine-sodium salt) 25 g/L, and DOPG (1,2-dioleoyl-sn-glycero3-phospho-(1′-rac-glycerol) (sodium salt)) 25 g/L, dissolved in chloroform, were purchased from Avanti Polar Lipids (Alabaster, AL), stored at −20 °C, and used without further purification. Calcium chloride was purchased from Sigma-Aldrich Inc. (Milwaukee, WI). BLES® (bovine lipid–protein extract surfactant) is a pulmonary surfactant containing charged and neutral lipids for use in the treatment of premature infants suffering from Neonatal Respiratory Distress Syndrome. The manufacturing process removes hydrophilic proteins, the majority of which is SP-A, and selects for hydrophobic phospholipids and SP-B and SP-C. The resulting lipid mixture is that found in natural surfactant which will include a low level of degraded/oxidized lipids. Previous work has elucidated BLES® phase behavior supplemented with 20 mol% cholesterol[63]. There is the formation of the expected

liquid ordered phase in vitro. However, in vivo studies show that the presence of physiological amounts of cholesterol has no effect on blood oxygenation levels or surface activity. BLES® was kindly provided by BLES Biochemicals Inc. as a dry film of approximately 100 mg of phospholipids from which a solution of 25 g/L was prepared in chloroform. Calcium chloride was purchased from Sigma Aldrich, Inc. (Milwaukee, WI) and used without purification. The fluorescent lipid Texas Red™ DHPE (1,2-dihexadecanoyl-sn-Glycero-3-phosphoethanolamine, triethylammonium salt) was purchased from Life Technologies (Grand Island, NY).

**Lipid film preparation**. All the samples for oxygen sensing, GISAXS, confocal microscopy and AFM analysis were prepared in the same way. Materials were mixed to obtain a final concentration of 25 mM in a final volume of 100 μL. The composition of the films was calculated in mole percentage, keeping the DPPC: DOPG molar ratio constant at 3:1. DPPC:DOPG lipid molar ratio was chosen to mimic terrestrial mammalian lung surfactant previously studied at the Leal group[17]. For BLES samples, the concentration was set at the effective lipid concentration for 3:1 DPPC:DOPG lipid samples. Cardiolipin was added to the diseased-mimetic films at 3 and 8 mol%, this was guided by literature values previously reported, e.g. 5–10 mol% of cardiolipin in human lungs affected by pneumonia is enough to impair surface activity[16]. Calcium chloride solution was prepared from fresh $CaCl_2$ dissolved in ethanol to obtain 0.09 M concentration and used without purification. Calcium chloride was mixed with the lipids at 4 mol% which is equivalent to 1 mM. The physiological concentration of ions and especially $Ca^{2+}$ in lung membranes can be complicated to measure accurately; however, most studies observe or use between 1 and 5 mM $Ca^{2+}$ [64–66]. Finally, the stock solutions were spin-coated directly onto the sensing device at 4000 rpm for 30 s. The resulting films were incubated at 45 °C for 48 h and then refrigerated until characterization at 4 °C. Since the pulmonary surfactant is a highly hydrated environment, and due to the lyotropic nature of the lipids in the mimic model and BLES systems, films were always kept at 98–100% RH during storage and characterization. We estimated the water layer in the films by X-ray, which is shown in Supplementary Fig. 10. The resultant $d$-spacing of healthy and diseased films are 9.52 nm ($d^{water}$ = 4.5 nm) and 9.58 nm ($d^{water}$ = 5.1 nm), respectively. In addition, by thermogravimetry, we found that the healthy and diseased films contain 14.9 wt% and 15.2 wt% of water, respectively, which is typical for DPPC-based systems[67].

**Sensor device preparation**. The first step for sensor fabrication is the transfer of the graphene layer to the $SiO_2$/Si wafer. Poly(methyl methacrylate) (PMMA) was spun on the graphene/copper foil as supporting layer. Copper foil was etched in copper etchant and a PMMA-coated graphene layer was supernatant. Subsequently, the graphene layer was cleaned in DI water and then transferred onto the $SiO_2$/Si substrate. The PMMA supporting layer was removed with acetone. A 3 nm-thick chromium/70-nm-thick gold composite layer was deposited by E-beam evaporator (Temescal) to form the metal electrodes. Cr/Au electrodes were photolithographically patterned by wet

etch-back processing using chromium and gold etchants. Graphene with channel-patterned photoresist (AZ 5214E) was dry-etched by reactive ion etching plasma (March CS-1701 reactive ion etcher) with 60 sccm $O_2$ gas flow and 50 W power for 60 s. After that, the remainder photoresist was removed by acetone. A schematic representation of this process is presented in Supplementary Fig. 1.

**Confocal fluorescence microscopy**. To evaluate the coverage and smoothness of the lipid layer adsorbed directly onto the device surface, confocal fluorescence microscopy experiments were performed by tagging the lipid solution with 0.1 mol% Texas Red$^{TM}$ DHPE dye. The morphology of lipid films on the sensor was monitored using a Zeiss LSM 800 confocal scanning laser microscope (Carl Zeiss AG, Germany) and a Plan-Apochromat 63X/1.40 Oil M27 objective lens. Supplementary Fig. 2 shows that the lipid coverage is excellent with virtually no defects detected in a micrometer lengthscale.

**Oxygen gas permeability measurements**. Measurements of oxygen gas permeability of the lipid films were performed using a temperature-controlled cryogenic vacuum probe station (LakeShore FWPX cryogenic Probe Station). The probe station was connected to oxygen and nitrogen gas tanks. Gas injection into the chamber for each species was controlled (Alicat Scientific, Inc.). Real time resistance measurements at >98% RH were conducted by the two-probe method with a drain bias of 0.1 V. The time consumed for saturation of the channel was measured as means to compare gas permeability.

**Grazing-incidence small-angle x-ray scattering—GISAXS**. GISAXS measurements were performed directly on devices covered with lung lipid films in-house (custom built with help of Forvis Technologies, CA, USA) and at the 12-ID-B synchrotron beamline at the Advanced Photon Source (APS), Argonne National Laboratory. The custom built in-house equipment is composed of a Xenocs GeniX3D Cu K$\alpha$ ultralow divergence X-ray source (8 keV), with a divergence of 1.3 mrad. Humidity and temperature control chambers were built by Forvis Technologies. At the APS 12-ID-B beamline, a 14 keV X-ray beam was focused on a $50 \times 10\ \mu m^2$ ($H \times V$) area at an incident angle of 0.05–0.2°. Pilatus2M (Dectris) detectors were used for GISAXS measurements both in-house and at APS 12-ID-B beamline. The sample-to-detector distance was calibrated using a silver behenate powder standard. Specular beam intensity was attenuated along the $z$ axis with a strip beam stop. Multiple measurements were carried on the same sample in a humidity chamber (RH > 98%) by varying the incidence angle to determine the most appropriate operating value. Both experiments at the APS and in-house were performed at room temperature and high RH (>98%). The measured scattering patterns were analyzed using the fit2D software package and an Igor-based package of tools for analysis of scattering data.

**Atomic force microscopy**. AFM measurements (MFP3D Asylum Research-Oxford Instruments) in humid air were employed to determine lung membrane film thickness by tip-scratching of well-defined sample regions[68]. To scratch the films we used monolithic silicon probes with spring constant of 40 N/m and nominal radius of 10 nm (Budget Sensors) in contact mode. The scan size was 5 μm with a width of 4:1. To make sure that only the film was scratched and the device (substrate) suffered no damage, a range of forces was applied to the film in order to find the maximum force that scratched the film but not the substrate. A loading force of approximately 3 μN was found to scratch only the films without damaging the substrate. Scratch tests using the same probes and applying the same force showed not to scratch the substrate. In order to ensure that the AFM tip reached the substrate, the films were scratched in the same spot during approximately 5 min. Subsequently, the scratched area was scanned using intermittent contact mode in the repulsive regime with the same type of tips. Depth of the scratches (i.e. sample thickness) was obtained by measuring height differences between the surface of the samples and the depth of the scratches through cross-section analysis. Supplementary Fig. 3 shows additional AFM probe-scratch measurements. AFM topography and phase imaging were also performed under high RH (>98%) in air directly on the devices covered with lipid films. The films were scanned in intermittent contact in the repulsive regime using silicon probes with spring constant of 40 N/m and nominal radius of 10 nm (Budget Sensors).

## Data availability
Data supporting the findings of this study are available from the corresponding authors upon request.

## Code availability
Codes associated with the findings of this study are available from the corresponding authors upon request.

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

## Acknowledgements

This work is funded by the Office of Naval Research (ONR) grant numbers N000141612886 and N000141812087 (DURIP-Defense University Research Instrumentation Program) and in part by the National Institutes of Health, grant number: 1DP2EB024377 (non-lamellar lipid structures). This research was carried out in part at the Materials Research Laboratory, University of Illinois. This work used resources of the Advanced Photon Source-beamline 12-ID-B, a U.S. Department of Energy (DOE) Office of Science User Facility operated for the DOE Office of Science by Argonne National Laboratory under contract no. DE-AC02-06CH11357. The authors would like to thank BLES Biochemicals Inc. for kindly providing us with the BLES used in all of our experiments involving lung surfactant extracts.

## Author contributions

C.L. designed the project. M.K. fabricated the sensor and measured oxygen permeation and GISAXS experiments. M.P.G. performed all AFM experiments. All authors analyzed the data and wrote the paper.

## Competing interests

The authors declare no competing interests.
