## [Peer Review File · Nature Communications]

Reviewers' Comments:

Reviewer #1:

Remarks to the Author:

The manuscript by Kim et al. summarizes data on an interesting study that assess permeability to oxygen of different lipid and lipid/protein preparations in the form of multilayers that simulate the composition and structure of pulmonary surfactant films, under normal and supposedly pathological conditions. The question of oxygen transport through the respiratory surface has been in my opinion largely underestimated and it is far of being understood. From this point of view, this study is highly relevant and of general interest. The authors have developed a novel graphite-based oxygen sensor able to measure oxygen concentration in minute membrane samples, which would be useful for study many other problems. This is, no doubt, a strength of the paper. However, many clarifications, and possibly some complementary experiments, are required before the full relevance of the paper can be assessed. This includes:

First, some more information is required on how the surfactant-mimicking films have been created. It is stated that the different materials have been deposited on the surface of the sensor by spin-coating of organic solvent solutions of lipids or lipid/protein mixtures. However, from a somehow homogeneous coating with lipid or lipid/protein molecules to true multilayered films in an aqueous environment (such as it is thought to be adopted by surfactant at the alveolar spaces), there is a distance. How the dry spin-coated films have been treated until getting the structure that is finally characterized? What is the level of hydration reached, and how it is ensured that it is comparable in all the different samples? Hydration is a major factor defining lipid polymorphism. In particular, non-lamellar phases such as those typically promoted by cardiolipin (CL) and Ca^{2+} are very lyotropic, particularly promoted under limited hydration. Limited hydration is not a factor expected at the alveolar surface.

Somehow connected with the previous point, the authors should be aware that BLES is not directly a native surfactant preparation. Although it is derived from natural resources, its composition (and therefore its structure) has been altered during production. For instance, a treatment of the organic extract of lipids and proteins obtained from bovine bronchoalveolar lavage (which is, by the action of the organic solvents, already far from the native original structure) is applied to remove cholesterol. As a result of that, the proportion of surfactant proteins could be also somehow reduced with respect to the original material. Reconstituting BLES back to multilayered aqueous suspensions end therefore in an structure that differs from that of native surfactant films, both compositionally and structurally.

A conclusion of the study is that the incorporation of high proportions of CL and Ca^{2+} induce formation of multiple intermembrane connections that end in a apparently higher rate of oxygen diffusion. This is used to conclude that such increase in oxygen permeation "can lead to an imbalance in alveoli gas exchange and gas composition in blood". It is difficult to believe that a mere increase in oxygen diffusion could create a pathological factor by itself, when in most cases impairment in oxygenation is a hallmark of lung injury. It could be possible that a non-lamellar phase, such as hexagonal HII, or cubic phase, is by itself a better oxygen conductive material. In this sense, the ability of CL and Ca^{2+} to promote negative curvature and formation of stacks is not surprising. The authors could use their setup to check for this possibility by testing a control material with a canonical mixture forming this type of structures (some of them typically contain CL and Ca^{2+}). If this is the case, by no mean that could indicate that a non-lamellar phase could be better for oxygen transport through the respiratory surface, because that would case many other problems, such as, for instance, poor mechanical resistance to compression, and low ability to reduce surface tension. On the other hand, respiratory pathologies are often accompanied by edema and fluid flow imbalance, leading to reduced concentration of surfactant and large water layers that would obviously interrupt oxygen diffusion. In this sense, it is obvious that the model studied here is far from being a good model of a pathological respiratory surface. It is surprising that nothing in this respect is mentioned at the discussion.

Why high Ca^{2+} is considered a pathogenic factor? Please provide references that it is actually the case. In this line, it is not clear which is the actual Ca^{2+} concentration used in these studies. Please, provide Ca^{2+} concentration in mM.

Another important question is that Ca^{2+} typically induces phase separation in membranes containing negatively-charged phospholipids such as CL. Could phase separation be also a oxygen diffusion-promoting factor? Additional controls in this direction could aid to assess a possible contribution in this respect.

Please, add some references on the first part of the introduction, when the main role of surfactant is introduced. In particular, the statement that pulmonary surfactant "is the first barrier against oxygen transport out of the alveoli into the bloodstream" should be documented. The dogma establishes that oxygen reaches the erythrocytes by free diffusion.

I do not agree with the statement that surfactant studies typically "reduce the system to a single monolayer of lipids and proteins spread thin on a pristine air/water interface". Although many relatively superficial studies, usually from fields far from pulmonology or respiratory biophysics, wrongly simplify surfactant to a DPPC monolayer, the authors could find many updated reviews firmly describing how surfactant films are highly cohesive multilayered structures, in which lipid/protein interactions play crucial roles. Please, incorporate some references in this line, because this is important for the readers to realize how pertinent is the study of preparations such as those proposed here.

In the paper by Olmeda et al. (reference 9 in the paper) it is clearly demonstrated that a minimal density of surfactant is required for oxygen diffusion rate to be reduced compared to that in surfactant-free water layers. This is in the line of the hypothesis opened by the authors at the introduction and should be acknowledged. Also, the fact that protein-promoted intermembrane connections are crucial for rapid oxygen diffusion. At the introduction, this question is documented by reference 4, which has nothing to do with intermembrane connection or oxygen diffusion, because the paper by Parra et al. actually establishes that hydrophobic surfactant proteins form defined pores into the membranes, with no clear connection at that point with oxygen diffusion.

The graphite sensor is composed of multiple graphite "bridges", through which oxygen concentration is measured. Is it correct that the values of oxygen concentration measured are somehow the average of all the measuring points? Could the structure of the sensor be used to assess the "homogeneity" of different regions of surfactant films to oxygen permeability? The calibration of the oxygen sensor has been made in free surfaces. The calibration should be repeated with a lipid film, to confirm that the range of response of the sensor does not varies in the presence of lipid films.

In Table 1, it does not make sense to put numbers of saturation times in the control experiments under both "control" and "BLES" columns. Control is not BLES by definition. The direct comparison of numbers from Table 1, lead apparently to think that "bridges" created by CL could somehow play the same role in oxygen diffusion than membrane interconnections based on proteins. It is even somehow proposed that proteins and CL could perhaps have an "additive" effect, because oxygen diffusion is faster in the presence of both elements. I do not think that the authors can directly conclude that from the data. For instance, protein bridges could be intrinsically different in the presence of CL, and be directly responsible of the fastest oxygen diffusion. Other alternative explanations could be equally possible.

In page 7, it is stated that "the lung system in its healthy state already exhibits inter-membrane contacts (or hemifusions) that are stabilized by lung proteins SP-B and SP-C [4]". This observation was proposed in reference #9, not 4. Alternatively, the authors may also like to make reference to any of excellent recent reviews describing in detail the role and action of hydrophobic surfactant proteins.

I am not so sure that the stalk phase diffraction peaks are only originated in the perfectly

orientated character of the multi-membrane structure. I think that the presence of conspicuous diffraction peaks mainly require regularly spaced membranes, which possibly only appear in the presence of the stalks.

The authors may like to know that the double set of lamellar peaks in the diffractogram of lung surfactant multilayered structures (both in whole native surfactant and in membranes reconstituted from organic extracts of surfactant similar to those studied here, was already described in Bernardino de la Serna et al. (Faraday Disc. 161, 535-548, 2013).

Please, what "topographic signatures of a tortuous membrane" (in page 8) does mean?

Phase images in AFM from lipid/protein structures could not only reflect differences in mechanical properties but also fundamental differences in material composition (i.e., lipid versus protein assemblies, with different response towards tip contact). Could the dots detected by a very different level of contrast at the phase images be constituted basically by protein, instead of the lipid touched by the AFM tip at most of the surface?

At the conclusions, when describing the different models in the control-like and disease-like samples, it would be important to acknowledge that the accelerated oxygen diffusion originated by the protein-promoted intermembrane connections in multilayered surfactant assemblies was already proposed by Olmeda et al (in reference 9), and demonstrated even in true native surfactant structures. This manuscript confirms that it is the case, using a totally different sensor and different preparations, which is very important and relevant. It also shows that changes in composition and possibly structure, as it could result in pathological contexts, have consequences on oxygen diffusion, which is an entirely novel observation. The work opens the novel idea that imbalanced oxygen diffusion could contribute significantly to respiratory pathologies as a consequence of the alteration of the composition and structure of the surfactant films. This may have important impacts on future basic and clinical research.

Reviewer #2:

Remarks to the Author:

In this manuscript, the authors developed high performance graphene-based FET sensors for the evaluation of oxygen gas permeability through supported lung membranes. Overall, the manuscript is very technological and sophisticated but well written. Many state-of-the-art techniques were used and developed; specifically, oxygen sensing, associated X-ray scattering and Atomic Force Microscopy structural characterization were used to identify the mammalian pulmonary membranes structural changes induced by cardiolipin and calcium ions. However, much of the data are rather preliminary, and the in vitro model of testing the graphene-based sensor is overly simplified in this manuscript, which preclude, in my opinion, publication in Nature Communication.

Major points:

There are many previous described graphene based oxygen sensor such as J. Phys. Chem. C2010114146610-6613 and Appl. Phys. Lett. 99, 243502 (2011), what's the technical innovation and advantages of the one developed here?

Although it is very cool to visualize how CL and Ca disrupt the membrane structure, the concept itself was a well described phenomenon. Also, what's the rationale of choosing 4 mol% Ca²⁺? The biggest issue I have with this manuscript is the oversimplified pulmonary membranes in healthy and diseased states. Simply put, authors only added CL and Calcium to bovine lipid-protein extracted surfactants to mimic the diseased pulmonary membranes. But, in real life, bacteria induced lung injury not only changes calcium balance and increases CL, but also affects many types of lipids and protein in surfactant. So, even for the proof of concept experiment to validate your graphene-based sensor, more studies are needed, e.g. use the real lipid protein extract from control and infected murine lung.

Reviewer #3:

Remarks to the Author:

Kim et al. presents a thorough comparison of oxygen transport in healthy and diseased models of pulmonary lipid membranes using an in-house engineered graphene-based oxygen sensor. Experiments are well designed with appropriate controls and the manuscript is very clear and well written. The work makes a significant contribution to the mechanistic understanding of oxygen transport in alveoli and is appropriate for publication in Nature Communications. Publication is recommended with only a few minor suggested changes to the text:

- 1) There appears to be a typo in the authors list. Is Mijung Kim supposed to have a marker for the Department of Electrical and Computer Engineering instead of Materials Science and Engineering?
- 2) Air/hydrogel interfaces are discussed in the introduction, but this does not appear to be relevant to the current manuscript. It is not clear how all of the model membrane systems, particularly the model healthy system, represent air/hydrogel interfaces, since there are no interconnected networks present. Also, hydrogel properties of these membranes are not assessed.
- 3) Some discussion of the known pathology of oxygen transport/gas exchange dysregulation under diseased conditions would be beneficial for the introduction. Is increased oxygen transport, as shown in the disease model data, a known characteristic of pneumonia? How does this contribute to the disease pathology? Furthermore, it is not clear how the tested CL and Ca²⁺ concentrations were selected for the experiments. Are these values based upon actual measurements from animals/patients with pneumonia or other diseases?
- 4) Although it is stated that lung extracts were provided by BLES Biochemicals Inc., there is insufficient information given in the methods for readers to reproduce these experiments. How exactly were the bovine lipids extracted? Importantly, how was the disease state induced and characterized for consistency? What disease state was used? Were these samples obtained from cows with a bovine respiratory disease, like BRD? Were the levels of distinct proteins and ions quantified between samples?

According to the comments of the Editor and the Reviewers, new data and data analysis was acquired, the manuscript was significantly restructured, and a comment (highlighted in a box) is provided below:

Reviewer #1: *The manuscript by Kim et al. summarizes data on an interesting study that assess permeability to oxygen of different lipid and lipid/protein preparations in the form of multilayers that simulate the composition and structure of pulmonary surfactant films, under normal and supposedly pathological conditions. The question of oxygen transport through the respiratory surface has been in my opinion largely underestimated and it is far from being understood. From this point of view, this study is highly relevant and of general interest. The authors have developed a novel graphite-based oxygen sensor able to measure oxygen concentration in minute membrane samples, which would be useful for study many other problems. This is, no doubt, a strength of the paper.*

However, many clarifications, and possibly some complementary experiments, are required before the full relevance of the paper can be assessed. This includes:

We would like to thank Reviewer #1 for acknowledging the importance and potential impact of our work. We are particularly thankful for their careful assessment and constructive criticism that guided us to produce a much-improved version of our initial manuscript. We conducted a significant amount of new experiments, data analysis, literature review, and restructuring of text and figures in order to address all the concerns. Please see below a point-by-point description of those revisions.

1. First, some more information is required on how the surfactant-mimicking films have been created. It is stated that the different materials have been deposited on the surface of the sensor by spin-coating of organic solvent solutions of lipids or lipid/protein mixtures. However, from a somehow homogeneous coating with lipid or lipid/protein molecules to true multilayered films in an aqueous environment (such as it is thought to be adopted by surfactant at the alveolar spaces), there is a distance. How the dry spin-coated films have been treated until getting the structure that is finally characterized? What is the level of hydration reached, and how it is ensured that it is comparable in all the different samples? Hydration is a major factor defining lipid polymorphism. In particular, non-lamellar phases such as those typically promoted by cardiolipin (CL) and Ca^{2+} are very lyotropic, particularly promoted under limited hydration. Limited hydration is not a factor expected at the alveolar surface.

We thank the reviewer for bringing up this important point, which of course we agree completely. After spin coating an homogeneous film of lipid and/or lipid-proteins, all samples were equilibrated at an environment of 98-100% relative humidity (RH) for at least three days before inspection. The humid environment was kept during all measurements (gas transport, X-ray scattering, AFM, and fluorescence microscopy). Under 100% RH the system is thermodynamically equivalent to bulk water (Chemical potential – $\mu = 1$). We understand this is not a dilute environment, but neither is the alveolar space. With the complex composition of cells, proteins, multilamellar bodies, tubular myelin, among others, the alveolar hypophase is a crowded environment where arguably the activity of water may effectively be even lower than 1. We are confident that studying lung membrane properties in a regime of near saturation relative humidity is representative of the environment in real systems (we add this comment now in page 2, 45-46 lines). We also measured the hydration level of the membranes and the results are equivalent to what is observed in fully hydrated DPPC-based systems (15-25 wt% of hydration water, Jürgens, E. Höhne, G. and Sackmann, E. (1983), “Calorimetric Study of the Dipalmitoylphosphatidylcholine/Water Phase Diagram” reference 74). We have now significantly extended this description in the materials and methods section (page 11, line 390, magenta text) and we report the measured weight fraction of water in the different systems to specifically address the reviewer’s concern and also to allow readers to fully reproduce our experiments.

2. Somehow connected with the previous point, the authors should be aware that BLES is not directly a native surfactant preparation. Although it is derived from natural resources, its composition (and therefore its structure) has been altered during production. For instance, a treatment of the organic extract of lipids and proteins obtained from bovine bronchoalveolar lavage (which is, by the action of the organic solvents, already far from the native original structure) is applied to remove cholesterol. As a result of that, the proportion of

surfactant proteins could be also somehow reduced with respect to the original material. Reconstituting BLES back to multilayered aqueous suspensions end therefore in an structure that differs from that of native surfactant films, both compositionally and structurally.

We thank the reviewer for this comment. We agree that we need to be very clear about the details of the BLES composition. Indeed BLES® is a natural lung surfactant obtained via a lavage method from freshly slaughtered animals. The manufacture process removes surfactant protein A, leaving a mixture of lipids and surfactant proteins B & C. The manufacturer stated to us that the lipid mixture at the end of manufacture is that found in natural surfactant which will include a low level of degraded / oxidized lipids. Nevertheless, previous work has elucidated BLES® phase behavior supplemented with 20 mol% cholesterol (Keating et al, Biophys. J. 2007, reference 70). There is the formation of the expected liquid ordered phase *in vitro*, however, *in vivo* studies show that the presence of physiological amounts of cholesterol has no effect on blood oxygenation levels or surface activity. These facts have been added to the manuscript in the section of “Materials” (page 12, line 364-68, magenta text).

3. A conclusion of the study is that the incorporation of high proportions of CL and Ca²⁺ induce formation of multiple intermembrane connections that end in a apparently higher rate of oxygen diffusion. This is used to conclude that such increase in oxygen permeation “can lead to an imbalance in alveoli gas exchange and gas composition in blood”. It is difficult to believe that a mere increase in oxygen diffusion could create a pathological factor by itself, when in most cases impairment in oxygenation is a hallmark of lung injury.

We are thankful that the reviewer brings up this point. We have to recognize that this fact is important and was superficially presented in the original manuscript. We have now modified the manuscript to include a discussion (and associated references) on the established phenomenon that increased oxygen supply in the lungs (hyperoxia) is a pathological factor critical in bacterial pneumonia. This has been observed in newborn mice where hyperoxia potentiates bacterial growth and inflammatory responses (Crouse DT et al, Infection and immunity, 1990, reference 36), as well as being an important cofactor for the development of acute lung injury and lethality in *L. pneumophila* pneumonia (Tateda K et al, The Journal of Immunology, 2003, reference 37]. This discussion is now added in page 3, line 89-92 and page 11, lines 341-344, magenta text.

4. It could be possible that a non-lamellar phase, such as hexagonal HII, or cubic phase, is by itself a better oxygen conductive material. In this sense, the ability of CL and Ca²⁺ to promote negative curvature and formation of stacks is not surprising. The authors could use their setup to check for this possibility by testing a control material with a canonical mixture forming this type of structures (some of them typically contain CL and Ca²⁺). If this is the case, by no mean that could indicate that a non-lamellar phase could be better for oxygen transport through the respiratory surface, because that would case many other problems, such as, for instance, poor mechanical resistance to compression, and low ability to reduce surface tension. On the other hand, respiratory pathologies are often accompanied by edema and fluid flow imbalance, leading to reduced concentration of surfactant and large water layers that would obviously interrupt oxygen diffusion. In this sense, it is obvious that the model studied here is far from being a good model of a pathological respiratory surface. It is surprising that nothing in this respect is mentioned at the discussion.

We agree completely that non-lamellar phases such as HII or bicontinuous cubic (QII) should be good O₂ conductive materials. HII is formed by water channels decorated by lipids where the alkyl chains form an hydrophobic continuum. In the case of QII, lipid bilayers are continuously folded in 3D (see schematic figure below) providing continuous aqueous and hydrophobic domains, the latter prone to O₂ permeation. In fact, stalk phases are related to these types of structures as they often appear as intermediate states (yet thermodynamically stable) before QII are formed (works of J. Seddon and T. Salditt cited throughout the manuscript). We have worked for years with QII phases. QII phases are robust and display strong mechanical resistance to compression.

We have followed the reviewer’s suggestion to measure the oxygen permeation through a canonical lipid material known to form QII phases. Glycerol monooleate (GMO) and 1,2-dioleoyl-3-trimethylammonium-propane (DOTAP) at 85:15 molar ratios form QII lipid films when equilibrated at 100% RH (see Kang et al. Adv Funct. Mater. 2016, reference 50). As shown in the X-ray diffraction pattern, GMO:DOTAP adsorbed onto the oxygen sensor device yields peaks that are completely consistent with a QII phase with the gyroid (Ia3d) symmetry. In this system, the measured O₂ permeation is strikingly similar (saturation time of 62 seconds) to that obtained for the stalk phase in the *diseased*-mimetic systems of our work. These results are now discussed in the manuscript in page 9, line 254-65, magenta text, and the figure below has been added to the supplementary information (SI) section of the paper (Fig. S9).

Figure S9. Oxygen permeation through non-lamellar lipid structures. (a) In-house GISAXS diffraction patterns of a well oriented bicontinuous cubic phase (gyroid) formed by the composition of glycerol monooleate – GMO, and the univalent cationic lipid 1,2-dioleoyl-3-trimethylammonium-propane chloride salt –DOTAP at a molar ratio GMO/DOTAP 85:15 at >98% relative humidity at 37 °C. (b) Oxygen permeation measurement through the cubic phase film – saturation time of $t_{\text{sat}} = 64$ s. (c) Schematic representation of a bicontinuous cubic gyroid. The gray surface represents the midplane of the lipid bilayer that separates two water domains (orange and blue)

Concerning the fact that “respiratory pathologies are often accompanied by edema and fluid flow imbalance, leading to reduced concentration of surfactant and large water layers that would obviously interrupt oxygen diffusion”. This is a very valid point. Interestingly hyperoxia is instead observed in mammalian lungs infected with pneumonia. We now added that fact on page 11, line 343-44, magenta text. It should be noted that in the event of lamellar swelling, the presence of stacks (membrane contacts) in intermembrane connections should retain O₂ permeation power. We used X-ray scattering to estimate the water layer thickness in all the films. The total d -spacing of *healthy* and *diseased* films are 9.52 nm ($d^{\text{water}} \sim 4.5$ nm) and 9.58 nm ($d^{\text{water}} \sim 5.1$ nm), respectively. In addition, by thermogravimetry, we found that the *healthy* and *diseased* films contain 14.9 wt % and 15.2 wt % of water, respectively. This data is shown in supplementary Fig. S10 and discussed in page 13, line 391-94, magenta text.

About the validity of our model system, we would like to reiterate that what inspired this work was the observation that lungs of mammals with respiratory diseases (such as pneumonia) show a significant increase in cardiolipin concentration (Ray et al, *Nature Medicine*, 2010, reference 38). It was speculated that this could affect lung surfactant phase behavior but this was not investigated until now. Our work shows that lung surfactant phase behavior will indeed most likely be affected and it may lead to enhanced oxygen permeation. Increasing oxygen flow is another symptom observed *in vivo* in mammals infected by pneumonia. Taken together, we argue that our model does provide an excellent platform to investigate the basic science behind such interesting mechanisms.

5. Why high Ca²⁺ is considered a pathogenic factor? Please provide references that it is actually the case. In this line, it is not clear which is the actual Ca²⁺ concentration used in these studies. Please, provide Ca²⁺ concentration in mM

That’s a valid point. The reason to add Ca²⁺ was to emulate the physiological concentration calcium (Ca²⁺) in lung membranes. However, it has been observed that elevated calcium levels in the lungs of cystic fibrosis patients might facilitate chronic behavior of *P. aruginosa* and it is postulated that it might affect the structure of the

surfactant in the alveoli (Broder et al 2016, reference 44). We have now added this note on section page 3, line 99-100, magenta text and also the actual mM concentration (1 mM) of calcium used to the Materials section.

6. *Another important question is that Ca^{2+} typically induces phase separation in membranes containing negatively-charged phospholipids such as CL. Could phase separation be also a oxygen diffusion-promoting factor? Additional controls in this direction could aid to assess a possible contribution in this respect*

We thank the reviewer for this comment. Indeed, Ca^{2+} typically induces changes in phase and structure of lipid membranes containing negatively charged phospholipids (De Kruijff et al, 1982, Pedersen et al, 2006 references 42 and 43) but usually at higher concentrations. We have used solid-state NMR (Steer et al, Langmuir 2018) to show that larger concentrations of calcium affects the order and dynamics of DPPG:DOPG membranes. This could of course also affect the permeation of O_2 . However, it should be noted that DPPC:DOPG (3:1) is at the cusp of phase separation even without any calcium and for those membranes the oxygen permeation is always very low compared to that of a stalk phase. We conducted a new control experiment with a DPPC:DOPG (4:1) membrane that definitely shows phase separation (Steer et al Langmuir 2018) and in fact the O_2 permeation ($t_{sat}=215$ s) is comparable but slightly higher than that of (3:1 DPPC:DOPG, $t_{sat}=200$ s). We add this discussion in page 10 lines 288-92 of the manuscript and Supplementary Fig. S6.

7. *Please, add some references on the first part of the introduction, when the main role of surfactant is introduced. In particular, the statement that pulmonary surfactant “is the first barrier against oxygen transport out of the alveoli into the bloodstream” should be documented. The dogma establishes that oxygen reaches the erythrocytes by free diffusion.*

Thank you for the recommendation. We added the following extra references in the Introduction, page 2, line 34-35, magenta text:

-Bastacky JA et al. “Alveolar lining layer is thin and continuous: low-temperature scanning electron microscopy of rat lung”. Journal of Applied Physiology. 79 (1995) 1615 (reference 3).

-Daniels CB et al. “Pulmonary surfactant: the key to the evolution of air breathing”. Physiology. 18 (2003) 151 (reference 2)

-Olmeda, Bárbara, et al. "Effect of hypoxia on lung gene expression and proteomic profile: insights into the pulmonary surfactant response." Journal of proteomics. 101 (2014) 179 (reference 11).

Traditionally, it has been thought that O_2 permeability through a lipid bilayer was as high as through the water layer of the same thickness. However, the permeability of oxygen through phospholipid membranes has been measured to be faster than through water (125.2 cm/s vs 60-80 cm/s - Subczynski et al. “Oxygen permeability of phosphatidylcholine-cholesterol membranes”. PNAS. 1989, reference 8).

8. *I do not agree with the statement that surfactant studies typically “reduce the system to a single monolayer of lipids and proteins spread thin on a pristine air/water interface”. Although many relatively superficial studies, usually from fields far from pulmonology or respiratory biophysics, wrongly simplify surfactant to a DPPC monolayer, the authors could find many updated reviews firmly describing how surfactant films are highly cohesive multilayered structures, in which lipid/protein interactions play crucial roles. Please, incorporate some references in this line, because this is important for the readers to realize how pertinent is the study of preparations such as those proposed here.*

Thank you for the recommendation. We agree. Our point was not to say that monolayer studies are “superficial”. Simply that monolayer/water systems cannot capture the complex phase behavior in the lung hypophase where multilayered lipid-protein complexes operate in a viscous (not much free water) environment. We have updated

the text to better describe this in section page 2, line 36-59, magenta text and included key review papers (reference 15) on the topic of lung surfactant multilayered systems in which lipid/protein interactions play crucial roles (REF #15: Perez-Gil J, Weaver TE. Pulmonary surfactant pathophysiology: current models and open questions. *Physiology*. 25 (2010) 132).

9. In the paper by Olmeda et al. (reference 9 in the paper) it is clearly demonstrated that a minimal density of surfactant is required for oxygen diffusion rate to be reduced compared to that in surfactant-free water layers. This is in the line of the hypothesis opened by the authors at the introduction and should be acknowledged.

Thank you for the recommendation. The following text was added to page 2, line 37-42, magenta text:

“Perez-Gil and co-workers (Olmeda et al, *Biochem. Biophys. Acta*. 2010 and Olmeda et al, *J. Proteomics* 2014) pioneered the study of the function of mammal pulmonary surfactants beyond lowering surface tension. One critical insight was that lung membrane structure could facilitate oxygen diffusion through the air-water interface. It was demonstrated that a minimal density of surfactant is required for oxygen diffusion rate to be reduced compared to that in surfactant--free water layers, suggesting that protein-mediated membrane contacts formed by pulmonary surfactant might have important properties to facilitate oxygenation through the thin water layer covering the respiratory surface.”

10. Also, the fact that protein-promoted intermembrane connections are crucial for rapid oxygen diffusion. At the introduction, this question is documented by reference 4, which has nothing to do with intermembrane connection or oxygen diffusion, because the paper by Parra et al. actually establishes that hydrophobic surfactant proteins form defined pores into the membranes, with no clear connection at that point with oxygen diffusion.

Thank you for detecting this imprecision. We corrected the reference that supports the intermembrane stalks related to oxygen diffusion with the following text on page 2, lines 57-59, magenta text:

“In addition, Perez-Gil and co-workers demonstrated that hydrophobic surfactant proteins SP-B and SP-C promote the formation of membrane contacts and that these proteo–lipid channels might play an important role in increasing membrane permeability and small molecule diffusion through the alveolar surfaces. Hence, akin to proteo–lipid pores, the inter–membrane contacts or stalks which are analogously hydrophobic are likely to facilitate oxygen gas transport through the alveolar membranes.”

11. The graphite sensor is composed of multiple graphite “bridges”, through which oxygen concentration is measured. Is it correct that the values of oxygen concentration measured are somehow the average of all the measuring points? Could the structure of the sensor be used to assess the “homogeneity” of different regions of surfactant films to oxygen permeability?

The calibration of the oxygen sensor has been made in free surfaces. The calibration should be repeated with a lipid film, to confirm that the range of response of the sensor does not varies in the presence of lipid films.

Yes, the extensive number of measuring spots was designed to acquire excellent statistics but one could envisage that the device is also used to probe local heterogeneities in a given system. However, we believe that this feature of the device should be explored in a separate effort as it would distract the main message of the current manuscript.

Thank you for the excellent suggestion of performing further calibration curves. The new calibration data shown below is now included in Fig. 2c.

12. In Table 1, it does not make sense to put numbers of saturation times in the control experiments under both “control” and “BLES” columns. Control is not BLES by definition.

Thank you for detecting this. Table 1 was changed: “Bear t_{sat} control = 20s” is now placed at the bottom of the table.

13. The direct comparison of numbers from Table 1, lead apparently to think that “bridges” created by CL could somehow play the same role in oxygen diffusion than membrane interconnections based on proteins. It is even somehow proposed that proteins and CL could perhaps have an “additive” effect, because oxygen diffusion is faster in the presence of both elements. I do not think that the authors can directly conclude that from the data. For instance, protein bridges could be intrinsically different in the presence of CL, and be directly responsible of the fastest oxygen diffusion. Other alternative explanations could be equally possible.

We thank the Reviewer for their comment. We agree that we do not know the form of the protein bridges in the presence of CL (this caveat is now included in page 9, lines 244-245, magenta text). However, we argue that it is difficult to find an explanation that is more consistent with the multiple data sets we acquired (device, scattering, AFM) combined with the rich body of literature. Specifically: 1) it’s clear that CL induces membrane stalks in model systems and that correlates with enhanced O₂ permeation. Not surprisingly, as membrane stalks are hydrophobic in nature, very much akin to the protein-induced “stalks”; 2) Protein-lipid systems (BLES) show higher O₂ permeation compared to protein-free systems; 3) Addition of CL to protein-lipid systems further enhance O₂ permeation.

14. In page 7, it is stated that “the lung system in its healthy state already exhibits inter-membrane contacts (or hemifusions) that are stabilized by lung proteins SP-B and SP-C [4]”. This observation was proposed in reference #9, not 4. Alternatively, the authors may also like to make reference to any of excellent recent reviews describing in detail the role and action of hydrophobic surfactant proteins.

We thank the reviewer for detecting this. We have made the correction in the manuscript (Olmeda et al 2010, instead of Parra et al 2013). We improved our reference list on this topic by adding a recent article describing the role of proteins in monolayer–bilayer transformations. [Baoukina S et al. Biochimica et Biophysica Acta (BBA)-Biomembranes. (2016), reference 61] as well as Parra E et al describing a “A combined action of pulmonary surfactant proteins SP-B and SP-C modulates permeability and dynamics of phospholipid membranes” in Biochemical Journal (2011), reference 62.

15. I am not so sure that the stalk phase diffraction peaks are only originated in the perfectly orientated character of the multi-membrane structure. I think that the presence of conspicuous diffraction peaks mainly require regularly spaced membranes, which possibly only appear in the presence of the stalks.

The authors may like to know that the double set of lamellar peaks in the diffractogram of lung surfactant multilayered structures (both in whole native surfactant and in membranes reconstituted from organic extracts

of surfactant similar to those studied here, was already described in Bernardino de la Serna et al. (*Faraday Disc.* 161, 535-548, 2013).

We changed the text in pages 7 and 9, lines 210-212 and 244-251, magenta text to clarify this fact. Multilamellar phases give rise to several $S(q)$ harmonics. When stalks form between layers, the $S(q)$ for the lamellar periodicity remains but an additional one arises coming from the orthorhombic unit cell of the stalks or membrane contact points (all papers from Salditt's group that first elucidated this structure in phospholipids are cited in the manuscript). This shows up in the form of diffraction points, not rings, as it is orientation dependent. As in, the stalks may always be present but just like unoriented lamellar phases show up as rings, unoriented orthorhombic stalks will also show up as rings and they would coincide with the rings of the lamellae and therefore be indistinguishable.

Thank you for the suggestion of the manuscript, which we admit we were not aware of. This is now referred to in relation to the discussion of Fig. 5. We cited here Bernardino de la Serna et al. (*Faraday Disc.* 161, 535-548, 2013, reference number 64.)

16. Please, what “topographic signatures of a tortuous membrane” (in page 8) does mean?

We are sorry for the confusion. We changed in the manuscript in page 6, lines 170-171, magenta text and now it reads : “It is noteworthy, that in addition to the oxygen permeability data, AFM results indicate that lung extract films have higher topographic roughness compared to the model system (Fig.3c). Those topographic features are significantly enhanced in the *diseased* state (Fig. 3d). This result will be further explored in the later sections of the manuscript.”

17. Phase images in AFM from lipid/protein structures could not only reflect differences in mechanical properties but also fundamental differences in material composition (i.e., lipid versus protein assemblies, with different response towards tip contact). Could the dots detected by a very different level of contrast at the phase images be constituted basically by protein, instead of the lipid touched by the AFM tip at most of the surface?

We thank the reviewer for this comment. We would like to clarify that the original AFM data shown in the manuscript was only performed on the lipid-only systems, so no proteins were present. However, we realize that for completeness and being able to clearly interpret differences between a lipid and a lipid-protein films we needed to execute AFM at the same conditions for all samples. In the new version of the manuscript (new Fig. 5) we include this new data (and some additional in Supplementary Fig. 11). We now show AFM data in two modes (height and phase) for lipid-only model systems and the lung extract in “healthy” and “diseased” mimetic states (acquired at 100 % RH). We show below just snippets of new Fig. 5 such as the height images for simplicity (the phase images align perfectly well with the interpretation of the results).

“Heathy state” model vs lung extract: the model lipid-only systems are smoother than the lung extract films.

“Diseased state” model vs lung extract: both model and lung extracts display clear “perforation-like” (indicated by the arrows) features consistent with the presence of membrane stalks. The phase data (in Fig. 5 g, h and

Supplementary Fig. 11) shows bright spots that align with perforations. This indicates that the AFM tip is in contact with a different “material” at the perforation sites, which is consistent with membrane pore-like “defects” of higher elasticity compared to the surrounding membrane. This fact is described in page 10, line 295-305.

18. At the conclusions, when describing the different models in the control-like and disease-like samples, it would be important to acknowledge that the accelerated oxygen diffusion originated by the protein-promoted intermembrane connections in multilayered surfactant assemblies was already proposed by Olmeda et al (in reference 9), and demonstrated even in true native surfactant structures. This manuscript confirms that it is the case, using a totally different sensor and different preparations, which is very important and relevant. It also shows that changes in composition and possibly structure, as it could result in pathological contexts, have consequences on oxygen diffusion, which is an entirely novel observation. The work opens the novel idea that imbalanced oxygen diffusion could contribute significantly to respiratory pathologies as a consequence of the alteration of the composition and structure of the surfactant films. This may have important impacts on future basic and clinical research

We thank the reviewer once again for acknowledging the importance of our work. As stated in page 11 line 336 we recognize that “Lung extract membranes are more permeable to oxygen gas compared to a lipid-only model systems, which agrees with other studies that show accelerated oxygen diffusion due to protein-promoted intermembrane connections in multilayered surfactant assemblies (Olmeda et al, 2010, reference 10)”

We now restructured the conclusions, page 10 to add the reviewer’s point:

“Cardiolipin induces the formation of periodic hemifusion contacts in lipid membranes, in particular in the presence of Ca^{2+} . It is likely that cardiolipin, which is over expressed in diseased lungs, promotes the formation of such stalks between bilayers of lung membranes in the alveoli. The formation of membrane stalks leads to enhanced oxygen gas permeability through the hydrophobic hemifusion points. Respiratory pathologies often cause edema and fluid flow imbalance in the alveoli and an interruption of oxygen flow should be expected, however, abnormally accelerated oxygen gas transport (hyperoxia) has been observed in mammalian lungs in diseased states, in particular during pneumonia (Crouse et al. 1990-reference 36, Tateda et al. 2003 – reference 37). We conjecture that this is consistent with a defective lung membrane with an increased number of stalk defects. The next stage of underpinning the role of cardiolipin and enhanced oxygen permeation during pneumonia will require the structural characterization and oxygen permeability of lung membranes extracted from healthy and diseased lungs of mammal model systems. At this point, our results raise new fundamental and conceptual insights on lung membrane function, indicating that changes in lung membrane structure and composition directly relate to oxygen permeation. This new observation can potentially enable innovative ideas for clinical research on the role of unbalanced oxygen diffusion through lung membranes in a pathological context.”

Reviewer #2: *In this manuscript, the authors developed high performance graphene-based FET sensors for the evaluation of oxygen gas permeability through supported lung membranes. Overall, the manuscript is very technological and sophisticated but well written. Many state-of-the-art techniques were used and developed; specifically, oxygen sensing, associated X-ray scattering and Atomic Force Microscopy structural characterization were used to identify the mammalian pulmonary membranes structural changes induced by cardiolipin and calcium ions. However, much of the data are rather preliminary, and the in vitro model of testing the graphene-based sensor is overly simplified in this manuscript, which preclude, in my opinion, publication in Nature Communication.*

We would like to thank the reviewer for recognizing the depth and soundness of our work. We hope that this much improved and restructured version of the manuscript, which contains a significant amount of new data and data analysis, is able to convince the reviewer of the importance of our findings. It is noteworthy that the strength of our work is much more well-articulated in this version. Specifically: this manuscript raises new fundamental and conceptual insights on lung membrane function, indicating that changes in lung membrane structure due to cardiolipin directly relate to enhanced oxygen permeation. This observation is potentially pertinent to the development of new clinical research as both increased cardiolipin and hyperoxia have been observed in multiple *in vivo* systems of mammalian lungs during bacterial pneumonia.

1. *There are many previous described graphene-based oxygen sensor such as J. Phys. Chem. C2010114146610-6613 and Appl. Phys. Lett. 99, 243502 (2011), what's the technical innovation and advantages of the one developed here?*

We agree with the reviewer's comment that there is a great body of literature on graphene-based oxygen sensors, many of which (including Chen et al, Appl. Phys. Lett. 2011) were cited in the original manuscript. We now also cite Joshi et al, J. Phys. Chem C 2010 (Reference 30). We do not intend to convey that our manuscript is a major technical innovation on oxygen sensors. Instead, we built on previous excellent developments (such as cited in the manuscript) and designed a sensor that enables us to measure with great accuracy oxygen permeation through model and extracted lung membranes at 100% relative humidity. We think that this type of sensor can be useful in other applications in which electronic devices are required to interface with biological membranes in appropriate environments of temperature and humidity. This point is now added in the manuscript in page 2, line 71-73, 84-86 in magenta text.

2. *Although it is very cool to visualize how CL and Ca disrupt the membrane structure, the concept itself was a well described phenomenon. Also, what's the rationale of choosing 4 mol% Ca²⁺?*

We thank the reviewer for this note. We agree that the structural effect of CL (and Ca²⁺) on lipid membranes has been studied before (our lab included). We do not claim this is a new concept. What is a new finding is that CL induces the formation of a periodic array (rhombohedral) of stalks in multilamellar constructs of lung membranes. Oxygen permeation through these stalks is significantly enhanced. Both of these findings are completely new. This associated to the fact that cardiolipin is significantly overexpressed in the alveoli of mammals infected with pneumonia and that hyperoxia (enhanced oxygen flow) is also observed *in vivo* on mammals afflicted with bacterial pneumonia, makes our conceptual work very relevant.

We admit that in the original form of the manuscript we did not properly justify the use of Ca²⁺ and its content. The reason to add Ca²⁺ was to emulate the physiological concentration of calcium in lung membranes. In addition, it has been observed that elevated calcium levels in the lungs of cystic fibrosis patients might facilitate chronic behavior of *P. aruginosa* and it is postulated that it might affect the structure of the surfactant in the alveoli (Broder et al 2017, reference 44). We have now added this note on page 3, line 99-100, magenta text and also the actual mM concentration (1 mM) of calcium used to the Materials section.

3. *The biggest issue I have with this manuscript is the oversimplified pulmonary membranes in healthy and diseased states. Simply put, authors only added CL and Calcium to bovine lipid-protein extracted surfactants to mimic the diseased pulmonary membranes. But, in real life, bacteria induced lung injury not only changes calcium balance and increases CL, but also affects many types of lipids and protein in surfactant. So, even for the proof of concept experiment to validate your graphene-based sensor, more studies are needed, e.g. use the real lipid protein extract from control and infected murine lung*

We thank the reviewer for this note. We agree that to completely correlate lung membrane structure to activity in real systems we must perform more work with *in vivo* models. We have modified the manuscript to be very clear about the potential impact of our work. In page 11, line 346-352, magenta text, we now state that:

“The next stage of underpinning the role of cardiolipin and enhanced oxygen permeation during pneumonia will require the characterization of structure and oxygen permeability of lung membranes extracted from healthy and diseased lungs of mammal model systems. At this point, our results raise new fundamental and conceptual insights on lung membrane function, indicating that changes in lung membrane structure and composition directly relate to oxygen permeation. This new observation can potentially enable innovative ideas for clinical research on the role of unbalanced oxygen diffusion through lung membranes in a pathological context”.

Nevertheless, we would like to make the note that all present pharmaceutical developments of lung surfactant replacement therapy build on analogous fundamental studies of lung surfactant phase behavior. It is important to further such basic-science studies and our work, together with other excellent developments from colleagues that are appropriately cited in this manuscript, is shifting the paradigm that the only function of lung surfactant is to lower the surface tension of the air/water interface.

Reviewer #3: *Kim et al. presents a thorough comparison of oxygen transport in healthy and diseased models of pulmonary lipid membranes using an in-house engineered graphene-based oxygen sensor. Experiments are well designed with appropriate controls and the manuscript is very clear and well written. The work makes a significant contribution to the mechanistic understanding of oxygen transport in alveoli and is appropriate for publication in NatureCommunications. Publication is recommended with only a few minor suggested changes to the text.*

We thank the reviewer for the constructive assessment of our work. We implemented all their suggested changes described below.

1. *There appears to be a typo in the authors list. Is Mijung Kim supposed to have a marker for the Department of Electrical and Computer Engineering instead of Materials Science and Engineering?*

Mijung Kim is indeed a PhD student at the Department of Electrical and Computer Engineering (ECE). Mijung started as a Materials Science and Engineering student when she initiated this work but has since switched to ECE.

2. *Air/hydrogel interfaces are discussed in the introduction, but this does not appear to be relevant to the current manuscript. It is not clear how all of the model membrane systems, particularly the model healthy system, represent air/hydrogel interfaces, since there are no interconnected networks present. Also, hydrogel properties of these membranes are not assessed.*

This is a very valid point. We corrected the manuscript in page page 2, line 46-52, magenta text to reflect this. What we wanted to convey is that many fundamental studies of lung membrane phase behavior focus on dilute systems on air/water interfaces. However, lung membranes are present in a rather viscous environment with not a lot of free water and the investigation of phase behavior in systems equilibrated at 100% relative humidity are thermodynamically more relevant. In future studies we will investigate air/hydrogel substrates but as the reviewer points out, this was not yet implemented in the current work.

3. Some discussion of the known pathology of oxygen transport/gas exchange dysregulation under diseased conditions would be beneficial for the introduction. Is increased oxygen transport, as shown in the disease model data, a known characteristic of pneumonia? How does this contribute to the disease pathology? Furthermore, it is not clear how the tested CL and Ca²⁺ concentrations were selected for the experiments. Are these values based upon actual measurements from animals/patients with pneumonia or other diseases?

We are thankful that the reviewer brings up this point. We have to recognize that this fact is important and was superficially presented in the original manuscript. We have now modified the manuscript to include a discussion (and associated references) on the established phenomenon that increased oxygen supply in the lungs (hyperoxia) is a pathological factor critical in bacterial pneumonia. This has been observed in newborn mice where hyperoxia potentiates bacterial growth and inflammatory responses (Crouse DT et al, *Infection and immunity*, 1990, reference 36), as well as being an important cofactor for the development of acute lung injury and lethality in *L. pneumophila pneumonia* (Tateda K et al, *The Journal of Immunology*, 2003, reference 37). We added this in page 3, lines 89-92, magenta text.

Cardiolipin concentration was chosen to match that detected by Ray et al, *Nature Medicine*, 2010, reference 38 in manuscript, in mammalian lungs infected by pneumonia. The reason to add Ca²⁺ was to emulate the physiological concentration of calcium in lung membranes. However, it has been observed that elevated calcium levels in the lungs of cystic fibrosis patients might facilitate chronic behavior of *P. aruginosa* and it is postulated that it might affect the structure of the surfactant in the alveoli (Broder et al 2017, reference 44). We have now added these references and notes on the introduction page 3, lines 96-100, magenta text and the actual mM concentration (1 mM) of calcium used to the Materials section.

Reviewers' Comments:

Reviewer #1:

Remarks to the Author:

In my opinion, the questions raised have been properly addressed in the revised version of the paper.

Reviewer #2:

Remarks to the Author:

In this revised manuscript, authors thoroughly examined/rebutted the critical issues I raised in the previous round of review. Some of the new findings were emphasized and better explained. Although the issue regarding using lung membrane from infectious mice remains unresolved, authors did clearly state the limitations and considered this as the future experiment. Considering the current focus of the manuscript, it is understandable that authors choose to test these in the future. Overall, the manuscript is technically sound and well-written and suitable for the journal in the current format.

Reviewer #3:

Remarks to the Author:

Authors have addressed all of my concerns with new text in the manuscript. In my opinion the manuscript is ready for publication in its current form.